# High annual-cycle repeatability suggests low flexibility to environmental changes in a near-threatened migratory shorebird

Philipp Schwemmer [1] ✉, Marie Donnez [2], Moritz Mercker[3], Stefan Garthe [1], Martin Boschert[4], Heinz Düttmann[5], Jaanus Elts [6], Thomas Fartmann [7,8], Wolfgang Fiedler [9,10], Frédéric Jiguet [11], Steffen Kämpfer [7], Michał Korniluk [12,13], Helmut Kruckenberg[14], Dominik Krupiński[15], Riho Marja[6,16], Markus Piha [17,18], Pierre Rousseau[19], Verena Rupprecht [20] & Pierrick Bocher[2]

Migratory species often repeat spatio-temporal patterns within their annual cycle. Although this may help to promote knowledge about local features and site quality, stereotyped behaviours may also create an ecological trap by preventing the flexibility required to adjust to environmental changes. Using a long-term international dataset, this study assesses 24 spatial and temporal parameters describing the repeatability of the entire migratory cycle in 94 individuals of the migratory near-threatened Eurasian curlew (*Numenius arquata*) that were tracked for up to 7 consecutive years using high-resolution GPS tags. Twenty-two parameters show significant repeatability, with the highest repeatability for use of the same breeding and wintering sites, indicating consistent faithfulness. All migration and stopover parameters during spring migration are also significantly repeatable, with lower repeatability for autumn migration, likely related to variable breeding success. The location of migration routes varies between consecutive years, but intra-individual similarity is significantly greater than inter-individual similarity. While the potential of adaptations to long-term environmental changes needs further studies (preferably including several cohorts of individuals) there are indications of a potentially maladaptive behaviour to short-term changes that should be carefully observed by site managers to conserve this near-threatened species.

Repeating the same spatio-temporal patterns each year may be beneficial for migrating species with a complex annual cycle, because using the same stopover, breeding and wintering sites promotes local knowledge about food availability, predation pressure and overall site quality[1,2]. However, retaining the same schedule and migration strategy and a high degree of site-faithfulness can create an ecological trap, given that a degree of flexibility is essential to allow the annual behaviour to adjust to changing anthropogenic or natural environmental pressures[1,3,4].

Regarding migrating birds, previous studies showed strong inter-individual differences in migration behaviour[5–7] as well as spatial patterns in breeding[8] and wintering areas[9], indicating that different individuals of the same species may display different spatio-temporal patterns. In contrast, intra-individual differences (i.e. repeated patterns displayed by the same individuals in a population), which require repeated observations of the same marked individuals during consecutive years, have been less well studied. High intra-individual site fidelity for breeding[10], wintering[11,12] and staging areas[4] has been reported for some species, but there are contrasting

results among species in terms of prospecting and vagrant behaviours to explore e.g., new breeding sites[13,14]. Previous studies assessing the repeatability of spatio-temporal patterns have largely concentrated on site-fidelity for breeding and/or wintering grounds, departure/arrival dates or parts of the migratory journey using satellite- or radio-telemetry, geolocators, transponders or colour ringing[6,12,15,16]; however, it is difficult to draw conclusions about intra-individual differences in spatio-temporal patterns for the entire annual cycle, which requires individual birds to be tracked across multiple years. The ongoing miniaturisation of global positioning system (GPS) techniques now allows us to assess repeatability across a bird's entire annual cycle, with high spatial and temporal resolution.

The Eurasian curlew (*Numenius arquata*, hereafter named curlew) is a medium- to long-distant migrant between West Africa up to the Arctic Circle[17,18] and is currently rated as near-threatened, showing a negative population trend across the East Atlantic Flyway[19,20]. Using an extensive international dataset, a previous study showed strong inter-individual differences in migration behaviour in curlews, suggesting

chain-migration along the East Atlantic Flyway, with individuals wintering farther south departing earlier and breeding farther south than individuals wintering farther north[21,22], as well as significant differences in timing of migration between the sexes, with females migrating earlier than males. There is thus currently good knowledge about inter-individual differences in timing of migration, use of stopover sites and connectivity between breeding and wintering sites in curlews; however, it is unclear if specific individuals behave the same way in consecutive years, i.e. the degree of intra-individual repeatability of migration patterns and fidelity for breeding and wintering grounds.

We therefore investigated intra-individual differences for curlews throughout their migratory cycle in terms of breeding/wintering ground fidelity, timing of migration and numbers of stopover sites, to draw conclusions about the degree of repeatability and to gain insights into the potential ecological flexibility of the species. Notably, the high-resolution of the dataset allowed us to question the similarity of migration tracks in subsequent years[23]. Our study used an extended long-term international dataset of high-resolution curlew GPS-tracking results comprising data for the same individuals for up to 7 consecutive years. We used 24 parameters such as repeated choice of breeding and wintering sites and a range of migration parameters, including the number of stopover sites, stopover durations, distances flown and similarity of migration routes in consecutive years. The long-term high-resolution tracking dataset provided an excellent resource for studying intra-individual repeatability throughout the entire migratory cycle of this species.

Our long-term GPS-tracking of the near-threatened Eurasian Curlew across eight years shows a high repeatability of its entire annual cycle and suggests a low potential for adaptation to environmental changes. Our

findings are discussed in the context of the potential need to adapt to future changing environmental conditions.

## Results

### Spatial and temporal patterns of curlews along the East Atlantic Flyway

The wintering distribution of tagged curlews ranged from the south-western Iberian Peninsula in the south to Scotland in the north, and from Ireland in the west to the Wadden Sea of Denmark and Germany in the east (Fig. 1). Breeding sites ranged from western France to areas north east of the Ural Mountains in Russia (Fig. 1). Migration was more condensed along the Atlantic and North Sea coasts, with the Wadden Sea being the most important stopover site (i.e. used by more than half of the tagged curlews), whereas migration patterns across the Baltic and further east showed a broad-front migration (Fig. 1). More details concerning curlew migration, with a focus on inter-individual differences, are available in Pederson et al.[21].

### Repeatability of spatial, migration and stopover patterns

Curlews showed a considerable degree of site-faithfulness to their breeding and wintering sites and repeatedly used the same sites in consecutive years, while their migration routes showed some plasticity (Fig. 2).

All repeatability values reported are adjusted to account for covariates including log-transformed track duration and cluster ID[24]. The latitude and longitude of the breeding (Fig. 3a, b) and wintering areas (Fig. 3c, d) were significantly repeatable among years, with $R$ values close to 1 (Supplementary Table S1).

Among the migration parameters, and in accordance with the high spatial repeatability in the use of breeding and wintering sites, the linear

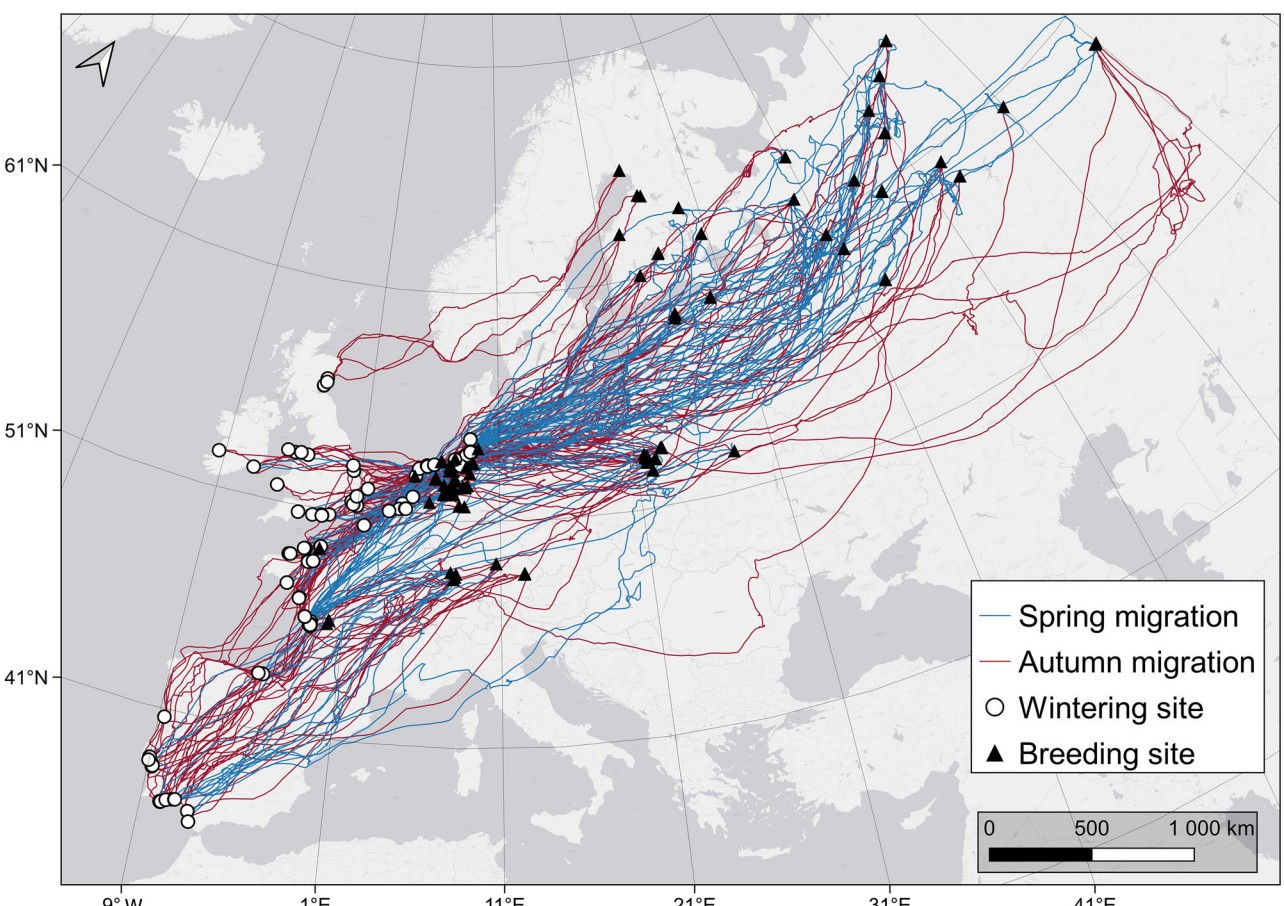

**Fig. 1 | Migration routes of curlews used in this study.** Spring (blue) and autumn (red) migration routes of 94 curlews tagged during the current study (between 2014 and 2021). White dots: wintering sites; black triangles: breeding sites. Source of basemap: Esri, HERE, Garmin, © OpenStreetMap contributors, and the GIS user community.

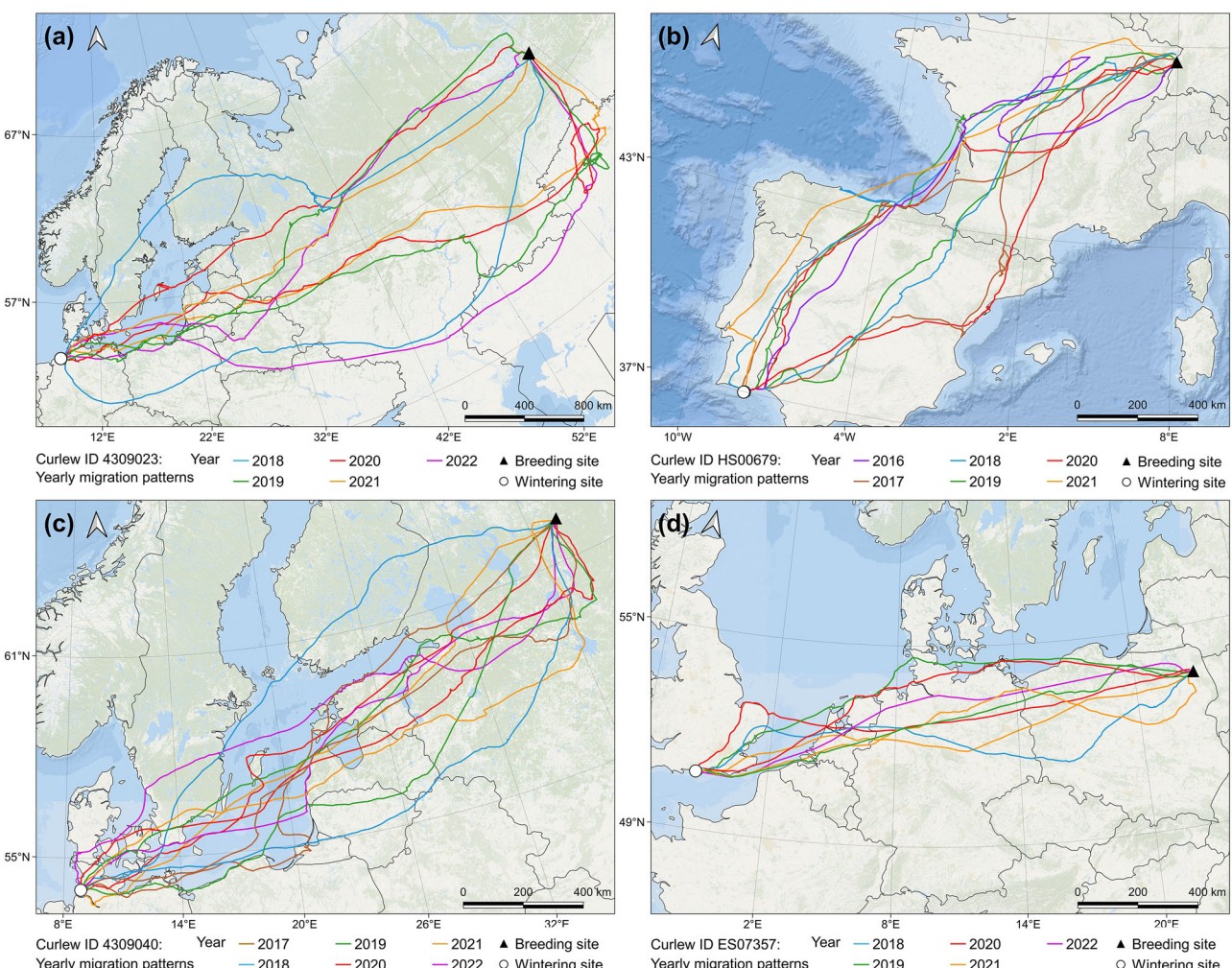

**Fig. 2 | Repeated migration of four curlews across multiple years.** Migration tracks of four individual curlews (**a-d**) over 5 to 6 consecutive years, respectively, (coloured lines) and use of same breeding (black triangles) and wintering sites (white dots). Source of basemap: Esri, GEBCO, NOAA, Garmin, and other contributors.

distance between breeding/wintering sites showed the highest $R$ value and highly significant repeatability (Fig. 4a; Supplementary Table S1), whereas the flown distance showed a much smaller $R$ value, although this was still highly significant (Fig. 4b; Supplementary Table S1). The $R$ values for the durations of spring (0.6) and autumn migration (0.4) were significant but showed weaker repeatability (Fig. 4c; Supplementary Table S1). Departure from the wintering site in spring was only moderately, but still significantly repeatable, whereas the repeatability of departure from the breeding site in autumn was not significant (Fig. 4d, Supplementary Table S1). The repeatability of arrival in the breeding site in spring was similar to that for the departure date, whereas the arrival in the wintering site after autumn migration showed low, but still significant repeatability (Fig. 4e, Supplementary Table S1).

Among the stopover parameters, mean and total stopover length had similar $R$ values, with significant repeatability for both spring and autumn migration (Fig. 5a, b). In contrast, the number of total stopovers was significantly repeatable during spring, but not during autumn (Fig. 5c; Supplementary Table S1).

Figure 6 gives a condensed overview of significant and non-significant $R$ values and their confidence intervals for each parameter of interest. Repeatability of nearly all migration and stopover parameters are lower for autumn than for spring migration.

Including sex and capture country as fixed effects did not materially affect repeatability estimates ($\Delta R < 0.02$ across all parameters). Neither covariate had consistent significant effects on migration timing or distance.

Weak tendencies for males to depart and arrive later in autumn did not alter the overall conclusions.

Sensitivity analyses using alternative buffer radii for winter and breeding clusters (150 km, 500 km) revealed highly consistent repeatability estimates between 150 km and 250 km buffers (r = 0.98, mean |$\Delta R$| = 0.035; Supplementary Fig. S1). At 500 km, the number of clusters decreased and several models became numerically unstable (R ≈ 0; Supplementary Fig. S1), confirming that overly large buffers reduce variance structure. We therefore base our inference on the robust 250 km results.

In a post-hoc analysis, we tested whether repeatability differed systematically among short-, medium-, and long-distance migrants. Across traits, we did not detect a consistent trend of repeatability with migratory distance in either spring or autumn (regression slope estimates close to zero with 95% confidence intervals overlapping zero; Supplementary Fig. S2).

## Track similarity

The overall mean and median distances between tracks in consecutive years did not differ among seasons (spring vs. autumn) (mean: z = 1.54, $p$ = 0.124; median: z = 0.36, $p$ = 0.717). In contrast the negative-binomial GAMs showed a highly significant effect for the parametric comparison of track type (same vs. different individuals) (mean: z = −4.61, $p$ < 0.001; median: z = −3.96, $p$ < 0.001), with a reduced mean distance of 25.9% and a reduced median distance of 28.5% for the level "same individual" (Fig. 7). This indicated that tracks for the same birds in consecutive years were more similar than tracks of different birds originating from the same spatial

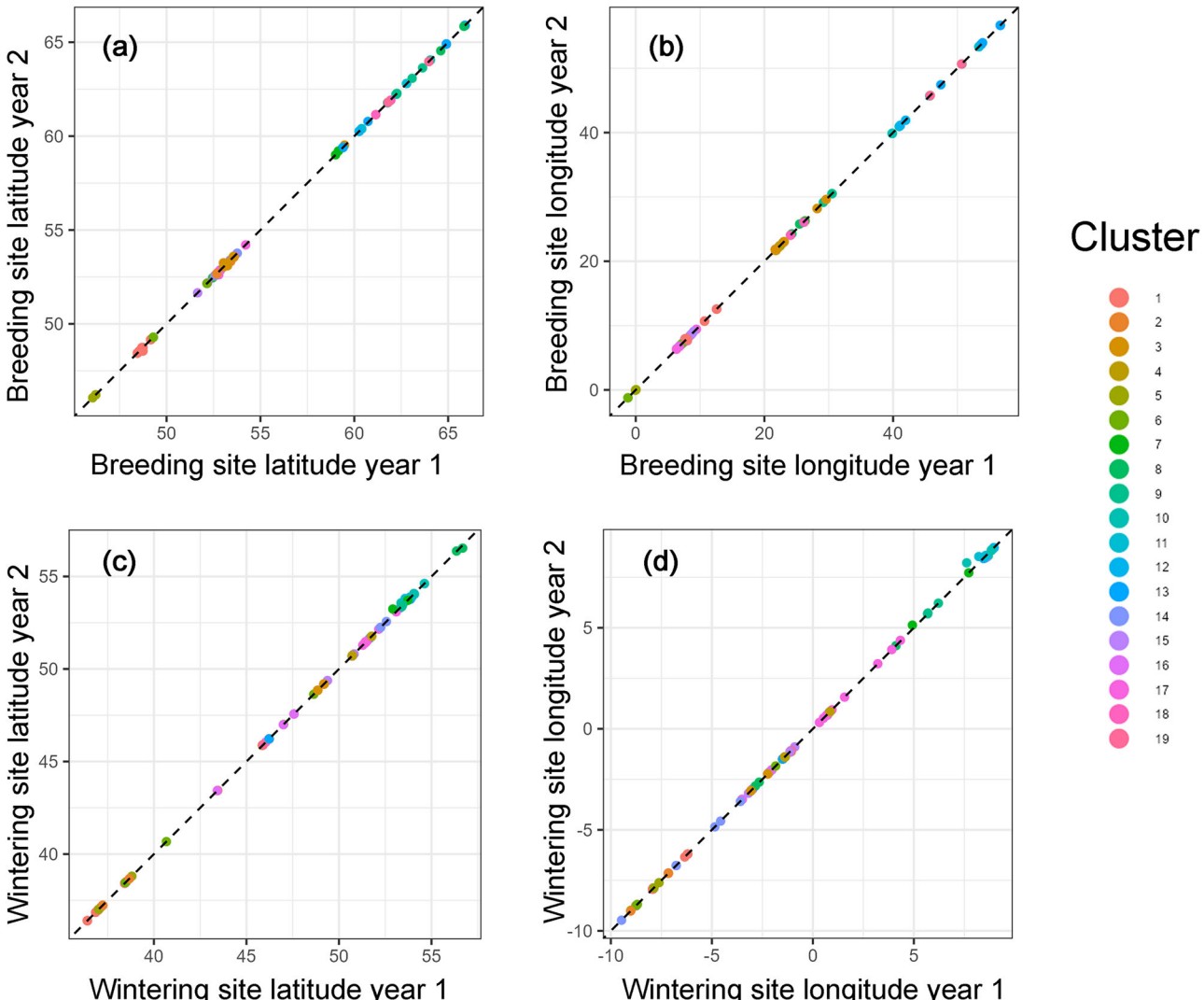

**Fig. 3 | Repeatability of spatial parameters of curlews among consecutive years.** Spatial repeatability of location of breeding (**a,b**) and wintering sites (**c,d**) of curlews during 2014-2021. Different colours depict birds breeding or wintering within the same spatial clusters, which were compared to avoid spatial biases (see Methods). Dashed line represents line of equality; overall variance indicated by scatter of points along the dashed line; variance for individual birds (degree of repeatability) indicated by distance of points from dashed line (closer to line, higher repeatability); and variance of the cluster ID indicated by different colours.

cluster, although mean and median distances showed a high variability as indicated by the large confidence intervals in Fig. 7. Parametric terms in these models are evaluated using Wald z-tests, which do not involve finite degrees of freedom (df → ∞).

## Discussion

The current study is one among the few tagging studies providing analysis of repeatability of a bird using data for a large number of individuals with high numbers of repeated observations (i.e., up to seven consecutive years) for the entire migratory cycle[16,25]. Our findings showed a high degree of repeatability throughout all stages of the migratory cycle for curlews, as a near-threatened species. Among the investigated parameters, spatial repeatability (i.e., fidelity of breeding and wintering sites) was the strongest, whereas temporal repeatability of migration and stopover parameters were still high (mostly highly significant), but were less repeatable than the use of breeding and wintering sites. Site-fidelity for breeding and wintering grounds is well-known among various other bird species[10–12]; however, curlews are known to have particularly small home ranges within their breeding[8] and wintering sites[9], and the current results suggest that different individuals confined themselves to these small sites in consecutive years. The combination of the small area used by curlews and the high repeatability of the use of this area might pose an important threat in terms of environmental or anthropogenic impacts. For instance, tagged breeding curlews exhibited constant use of nest sites in areas of changing agricultural practice and reforestation in western Russia[8].

Among the migration parameters, the linear distance between breeding and wintering grounds showed the highest $R$ value, in keeping with the high intra-individual site fidelity for breeding and wintering grounds. Most other migration and stopover parameters were still repeatable, but with lower $R$ values. Notably, the departure and arrival dates were less repeatable than expected based on a previous study of curlews wintering in the Wadden Sea, which showed highly repeatable departure dates from the wintering ground[26], suggesting that the location of the breeding or wintering cluster had a greater effect on the departure/arrival dates than the repeatability. This is in line with strong inter-individual differences in departure/arrival dates of birds wintering/breeding in different regions along the East Atlantic Flyway, which exhibit chain migration[21], with birds wintering further north breeding further north and those wintering further south breeding further south. In addition, long-distance migrants were shown to exhibit less variation in the timing of their migration compared with short-distance migrants[27,28]. However, after separating the birds used in this study into short, medium and long-distance migrants, we were not able to detect any

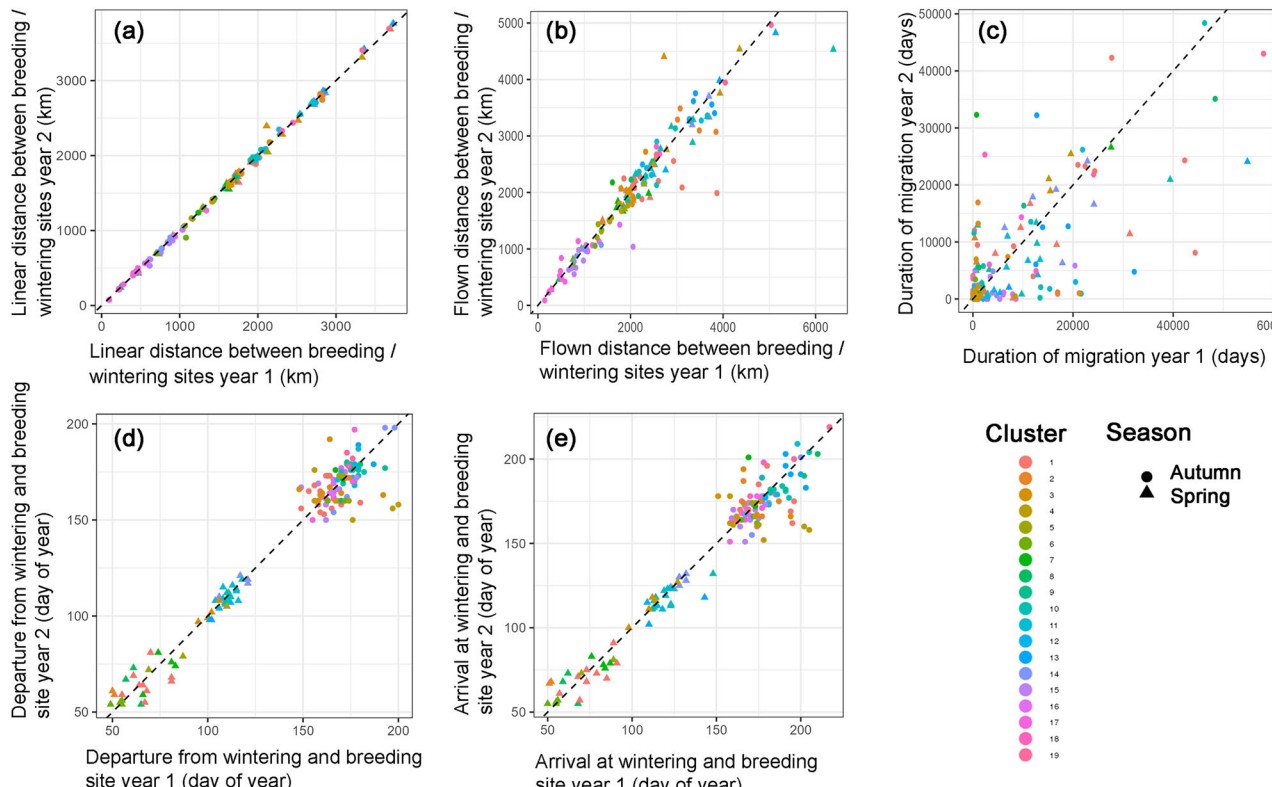

**Fig. 4 | Repeatability of curlew migration among consecutive years.** Repeatability of migration parameters of curlews during 2014-2021. Linear distance (**a**) and flown distance (**b**) between breeding/wintering sites, duration of migration (**c**), departure from wintering and breeding sites (**d**), arrival at wintering and breeding sites (**e**). Dots: autumn migration; triangles: spring migration. See Fig. 3 for further descriptions of the plots.

differences in repeatability values of any spatial, migration or stopover parameter used in this study among the three migration classes.

Although most of the tested migration and stopover parameters were highly significant for both migration seasons, the $R$ values were always lower for autumn than for spring, and notably, the repeatability of the departure date from the breeding ground was not significant. This is in line with previous studies on other taxa of birds[29], including the taxon of Numeniini[25,30], and the current study now proving it for curlews. This was probably due to differences in breeding success between consecutive years, with unsuccessful curlews leaving the breeding sites earlier than successful ones[22]. Males also tend to stay at the breeding site longer than females[21,31]; however, given that the sex ratio among our tagged birds was nearly equal and we compared individual birds, this should not affect the repeatability of the departure date. Although the departure from the breeding grounds was not significantly repeatable, the arrival at the wintering sites was repeatable, suggesting that curlews adjusted the timing of their migration to arrive at relatively the same time each year. Previous studies suggest that a high degree of site-faithfulness in wintering areas is beneficial for birds to make use of predictive prey resources, lower predation risks, and to be able to start with the postnuptial moult in time[32,33]. A repeated arrival pattern could enhance these benefits.

In addition to breeding success, other environmental factors, such as weather conditions and food availability at the breeding sites[34,35] may cause low repeatability in the timing of migration and stopover parameters. In line with the current results, however, a previous study found no strong effect of weather conditions on the repeatability of departure dates of curlews from a wintering site in the Wadden Sea[26], and suggested the existence of a strong genetic component triggering the timing of the migration. There is also some indication that curlews fine-tune their migration according to the previous year's green-up date in sub-Arctic and Arctic breeding sites[36], indicating their ability to adjust to environmental parameters if needed.

While several decades ago winter movements of curlews to areas further south have been described based on ring recoveries[37], there were only two individuals in our dataset which showed sudden movements as a consequence of a short-term cold spell[38]. This might either reflect missing flexibility or reflect a decrease in the frequency or severity of cold spells. However, these were the only documented observations of this behaviour among the 94 individuals in the dataset (i.e., 4.3% of the birds). Although short-term cold spells during the years of our study may have decreased in frequency or severity to decades ago, curlews (among other shorebirds) have been described to die from sudden cold spells in their wintering grounds, which might likely be a consequence of their high site fidelity[39].

A previous study on black-tailed godwits (*Limosa limosa*) showed that, although individuals may not respond to a high degree to environmental changes, long-term observations of some bird populations suggested some adaptation[40], possibly as a result of new recruits to the population showing different behaviours[41,42]. If the same mechanisms apply to curlews, individual adult curlews might lack the ability to respond to environmental pressures, whereas the population as a whole might adapt sufficiently to changes.

The linear distance between wintering and breeding sites was highly repeatable, in line with the high spatial repeatability of individual site fidelity for wintering and breeding grounds. As expected, the $R$ values for flown distances were much smaller than those for linear distance, because curlews chose slightly different migration routes in consecutive years when commuting between wintering and breeding sites. Nevertheless, the locations of migration tracks for individual curlews showed a significantly higher similarity than the tracks of other individuals from the same breeding or wintering cluster, although the similarity showed a broad confidence interval. Like other shorebird species, curlews migrate in small flocks, at least just after departure[43], but then separate from conspecifics along the route to arrive at their known breeding/wintering grounds. Curlews depart with the flock as a whole likely making a decision about the exact migration, possibly

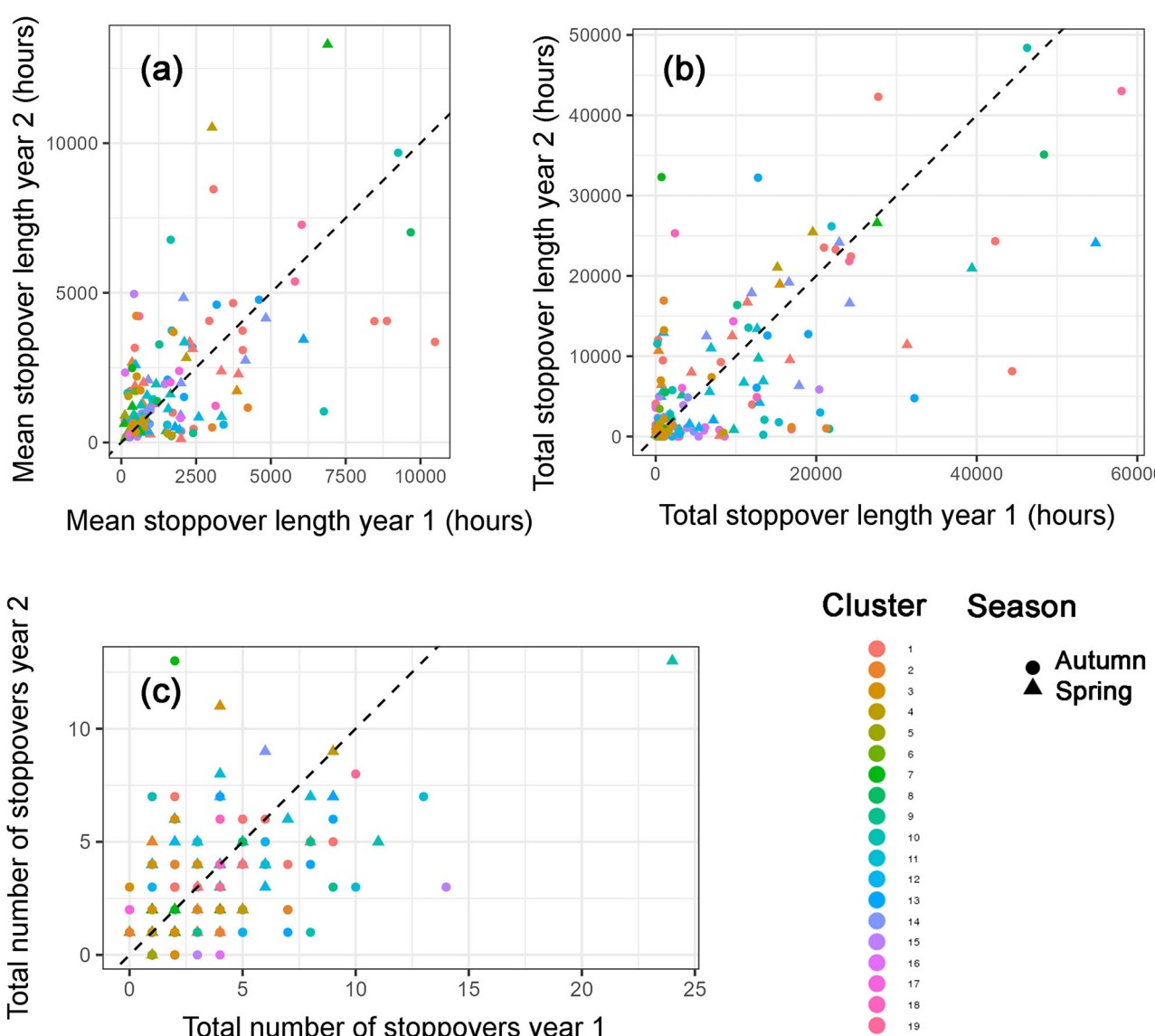

**Fig. 5 | Repeatability of stopovers of curlews among consecutive years.** Repeatability of stopover parameters of curlews during 2014–2021. Mean stoppover length (**a**), total stoppover length (**b**) and number of total stoppovers (**c**). Dots: autumn migration; triangles: spring migration. See Fig. 3 for further descriptions of the plots.

also considering wind assistance[3], leading migration routes of single individuals to differ slightly from year to year. The start and end of migration are thus more likely to resemble the tracks from preceding years, given that their high site fidelity means that curlews will select the same breeding and wintering sites and thus take similar routes, at least at the start and end of their journeys.

Other closely related species from the taxon of the Numeniini from areas elsewhere are also known to show high repeatability, e.g. the departure from breeding and wintering grounds and timing of moult in bar-tailed godwits (*L. limosa baueri*) from New Zealand[25]. However, most other species from this taxon obviously do not exhibit high repeatability throughout their entire annual cycle. For example, Marbled godwits (*L. fedoa beringiae*) from Alaska showed high site fidelity to breeding and wintering sites, but exhibited high individual flexibility in timing of migratory movements[44], and Alaska and Iceland-breeding whimbrels (*Numenius hudsonicus* and *N. phaeopus*) showed a high repeatability in the timing of spring migration, however, exhibited greater variation in other migratory traits or timing of breeding[30,45]. In contrast, our study showed a surprisingly high repeatability in all annual phases of curlews.

For the interpretation of the results, it is also important to discuss the nature of the repeatability measures in general. Previous studies pointed

out that measurement errors, e.g. from spatially imprecise radio transmitters might lead to biases in the repeatability estimates[46]. This is, however, not the case in our study as we consistently applied the same type of high-resolution GPS-devices for all individuals used. Furthermore, repeatability estimates quantify the proportion of total variance attributable to consistent differences among individuals, but they do not describe the absolute magnitude of within-individual variation[25]. Accordingly, high repeatability does not imply low intra-individual variability, but rather that among-individual differences dominate relative to within-individual variation. In our study, repeatabilities were estimated from variance components of mixed-effects models, explicitly accounting for both inter- and intra-individual variance. We therefore interpret repeatability as a relative measure of consistent individual differences, not as a measure of absolute consistency or as direct evidence for stable personality traits (see also Dingemanse et al.[46];Stuber et al.[47]). We deliberately did not show results of inter-individual variances as this was reported in great detail in a previous study on curlews[21].

This study provided a detailed investigation of the intra-individual repeatability of spatio-temporal patterns throughout the entire migratory cycle in a migratory bird species. The results highlight the high repeatability of the entire migratory cycle in the near-threatened curlew. This suggests a

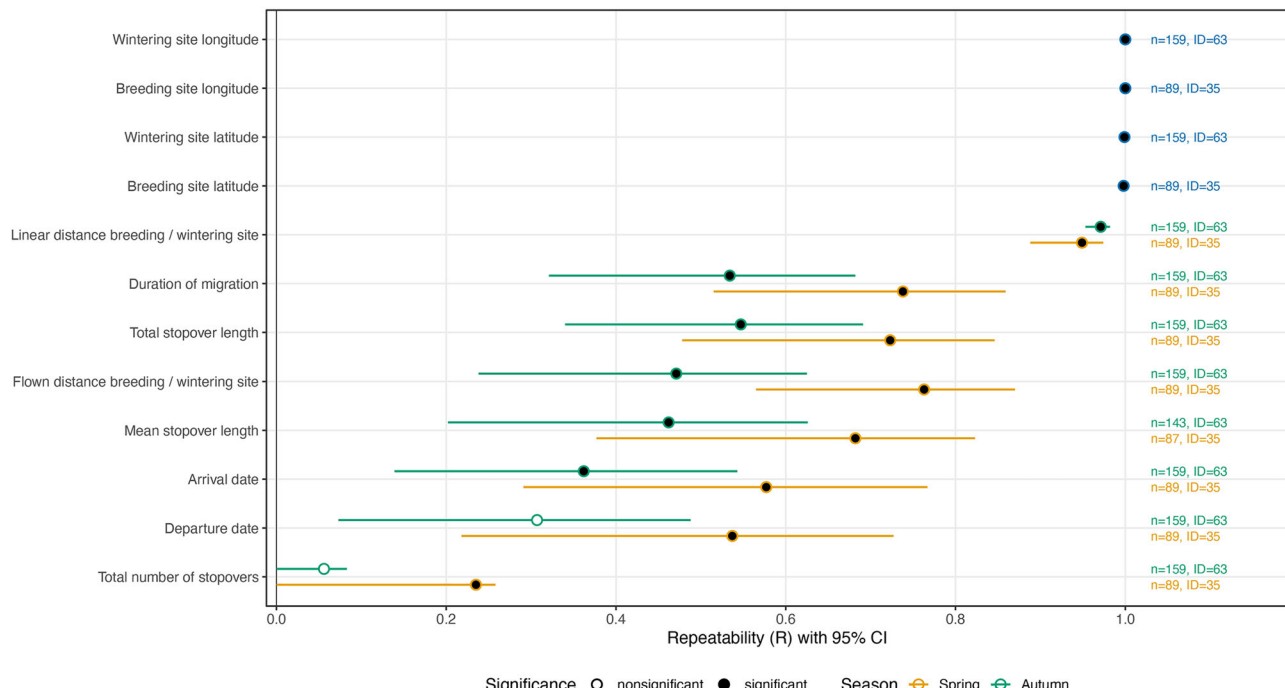

**Fig. 6 | Repeatability estimates of spatial, migration and stopover parameters.** R values of curlews during 2014-2021 (dots) with their corresponding 95% confidence intervals (bars) for each considered parameter during both seasons (spring: orange; autumn: green). Black dots: significant R values; open dots: insignificant R values; n: number of repeated observations; ID: number of birds. For concrete R values and confidence intervals see Supplementary Table S1.

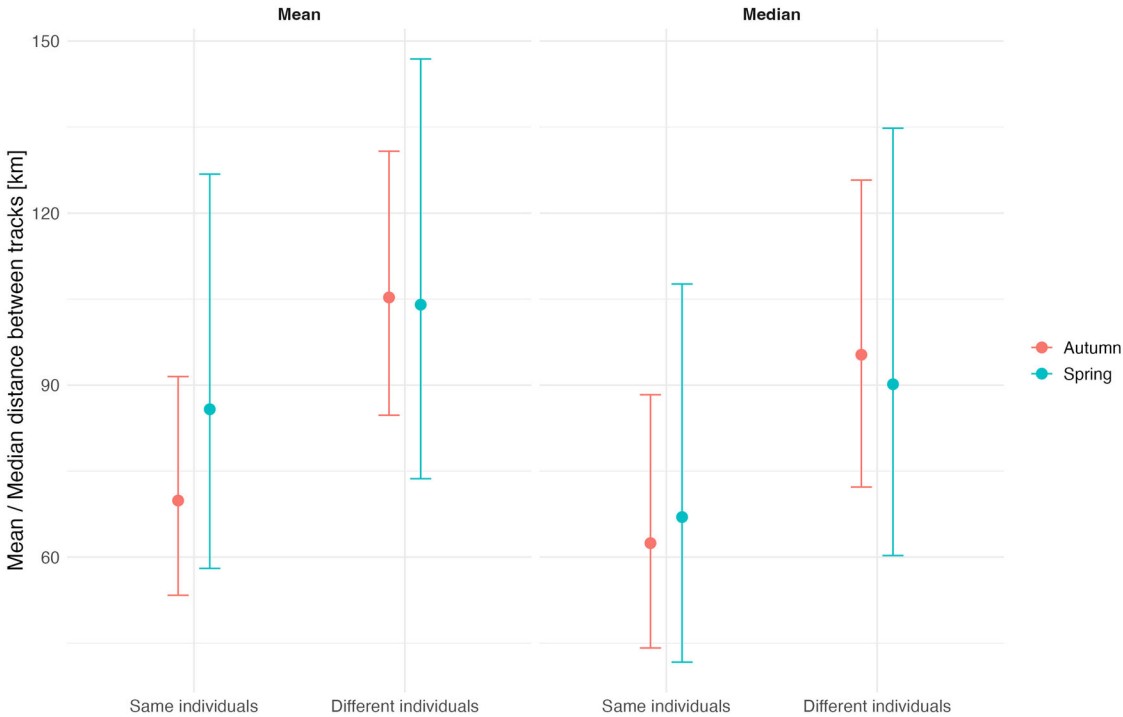

**Fig. 7 | Similarity of chosen migration tracks of curlews.** Similarity of migration tracks of curlews during 2014-2021 during autumn (red) and spring (blue) expressed as mean overall distance and median overall distance of tracks for the same and different individuals originating from the same spatial clusters in consecutive years (see Supplementary Fig. S3). Dots: mean or median values; bars: 95% confidence intervals (CIs). Means for spring: 104.0 km, 95%CI 73.7–146.9 km (different individuals) vs. 85.8 km, 95%CI 58.0–126.8 km (same individuals); means for autumn: 105.3 km, 95%CI 84.7–130.8 km (different individuals) vs. 69.9 km, 95%CI 53.4–91.5 km (same individuals). Medians for spring: 90.1 km, 95%CI 60.3–134.8 km (different individuals) vs. 67.0 km, 95%CI 41.7–107.6 km (same individuals); medians for autumn: 95.3 km, 95%CI 72.2–125.8 km (different individuals) vs. 62.4 km, 95%CI 44.2–88.3 km (same individuals).

low flexibility of curlews to react to changing environmental conditions, by maintaining their annual routine irrespective of the environmental parameters[26]. Despite this apparent lack of flexibility, however, it is possible that curlews might be able to adjust to environmental changes if the environmental stressors become strong enough, as found for black-tailed godwits[12,48] and bar-tailed godwits (*L. lapponica*)[49], thus avoiding the ecological traps described for other species[4,48,50,51]. The strong repeatability found in the current study might thus indicate the absence of strong environmental pressures for curlews or the lack of ability to adjust to such changes. Wintering sites in western Europe are comparatively well protected against anthropogenic impacts (in contrast to the loss of tidal flats in the Yellow Sea[52]). However, climate change induced effects such as sea level rise are also impacting roosting and foraging sites in the wintering grounds of protected areas[53]. These environmental changes act over larger time spans that cannot be resolved within the time period covered by this study. In contrast, the cases of weather-induced mortality within wintering sites[39] as well as effects of afforestation and changes of agricultural practices in northwest Russian breeding sites[8] illustrate likely effects of missing flexibility to shorter-term environmental changes in curlews. The question about the adaptive potential of curlews to react to long-term environmental changes cannot easily be solved without long-term studies. Given that curlews have a high annual survival and relatively slow life history strategy, high repeatability can be an adaptive response to variability in environmental conditions if resource availability does not reach critical thresholds[54,55]. Furthermore, previous studies found that birds were able to adapt to environmental changes by adjusting parts of their annual cycle[30]. However, as the repeatability exhibited by curlews is constant across their entire migratory cycle, there is currently no indication that individuals of the same cohort are responding to longer-term changes. Prior studies of black-tailed godwits demonstrated that juveniles could exhibit new migration routes within one generation[41,42]. Given that the current dataset only comprised adults, further studies including tagging juveniles are needed to determine if this might also be the case for curlews, and to assess the degree of plasticity to long-term environmental changes of the population as a whole (see also discussion in Carneiro et al.[30]). Furthermore, fine-scale analyses of the repeated use of habitats can shed light on potential impacts of short-term environmental changes on curlews. This would inform conservation management about the degree of flexibility of individuals from the same cohort. Therefore, future work should investigate the repeatability of home ranges in wintering and breeding grounds and thus explore the flexibility in habitat choice. We also did not yet analyse the repeated use of stopover sites. For this, it would be necessary to include remote sensing data to account for habitat changes among years, exceeding the scope of the current study. However, the high variation in mean distance of GPS fixes of migration tracks in consecutive years suggests that curlews might select different stopover sites, and a follow-up study should analyse the repeated use of these sites and their connectivity. Together with the current results, these future studies will help to assess the environmental flexibility of curlews during migration and thus inform conservation management.

Although the current results cannot determine if curlews lack the flexibility to adapt to longer-term environmental changes, the study clearly highlights the risk of potential maladaptive behaviour in particular with respect to short-term changes, given the extremely high repeatability throughout the migratory cycle of this near-threatened species. Thus, conservation management should aim to carefully monitor the effects of anthropogenic impacts and short-term environmental changes on curlews.

## Methods
### Study area and tagging of curlews
The study area covered most of the breeding and wintering ranges of curlews along the East Atlantic Flyway from north-east Russia to the Iberian Peninsula[17] (Fig. 1). We caught a total of 94 adult curlews (44 males and 50 females) between 2014 and 2021 across different programmes in Germany

**Table 1 | Numbers of tagged curlews used to assess the given number of repeated spring and autumn migrations and breeding and wintering periods during 2014-2021**

| No of repetitions | Number of birds used for repeatability of… | | | |
| --- | --- | --- | --- | --- |
| | …spring migrations | …autumn migrations | …breeding periods | …wintering periods |
| 2 | 30 | 46 | 62 | 59 |
| 3 | 2 | 11 | 18 | 24 |
| 4 | 5 | 6 | 8 | 3 |
| 5 | 2 | 3 | 3 | 6 |
| 6 | 1 | 1 | 2 | 1 |
| 7 | 0 | 0 | 0 | 1 |
| Total | 40 | 67 | 93 | 94 |

(n = 63), France (n = 25) and Poland (n = 6), either on the nest during incubation using cage traps, clap nets and scoop nets (n = 69), or at their wintering roosts using mist nets (n = 25). Immature birds were not caught because of the chance that they might show different behaviours in the first years. All birds were equipped with solar-powered GPS-Global System for Mobile Communication (GSM) data loggers that were attached using leg-loop (n = 30)[56] or wing-loop harnesses (n = 64)[57] with silicon or Teflon tape. The following device types were used: (1) GSM Radio Tag M9 (16 g; Milsar, Poland), (2) OrniTrack 10 (10 g), OrniTrack E10 (12 g), OrniTrack 15 (15 g), OrniTrack 20 (20 g) (all Ornitela, Lithuania), (3) Sterna VHF SRD (7.5 g), Saker-L (16 g), Skua (17 g) (all Ecotone, Poland) and (4) Solar GPS (14.5 g; e-obs, Germany). Prior to tagging, all birds were weighed to the nearest gram and it was confirmed that the tag weight did not exceed 4% of the body mass of the birds, in line with previous studies, to avoid device effects[58]. Curlews were sexed either morphologically (after Summers et al.[59].) or genetically using feather samples (Tauros Diagnostics, Berlin, Germany). Birds were kept in darkened tents until deployment of GPS devices in order to reduce stress.

According to battery stage, the devices recorded time of day (UTC), geographical position and flight speed at intervals between 1 and 1,440 min (99.6% of intervals <60 min). Data were transferred either via the GSM network to a central server or via very high frequency (VHF) to a base station placed at the wintering site and eventually stored in the data portal www.movebank.org[60].

### Parameters investigated for intra-individual repeatability among years
Spatial and temporal repeatability among successive years was assessed for the entire annual cycle of the curlews. Data were only included for curlews with a minimum of two complete spring or autumn migrations, respectively, and/or with a minimum of two wintering or breeding periods (maximum: seven repetitions). Seventy-six individuals were available for analyses of repeatability of migration tracks (40 individuals spring, 67 autumn), 94 individuals were available for repeatability of breeding and 93 for wintering periods (Table 1). To account for different fix intervals of the tags, we discarded the complete tracks of individuals with gaps of > 24 h (following Pederson et al.[21]).

Prior to the analyses, we defined if GPS fixes of a bird were recorded at breeding, wintering or stopover sites. For this, we followed the method described in Page et al.[61]. and Donnez et al[9]: (1) breeding sites: all fixes in the vicinity of the nest site after the northbound movement during spring migration and before the southbound movement during autumn migration; (2) stop-over sites: all fixes with slow and non-directional movements located between breeding and wintering sites and separated by in-flight positions with high speeds and directional movements; (3) wintering sites: southernmost sites where curlews stayed >20 d (see Donnez et al.[9]), southbound migration during autumn ended and northbound migration

during spring began. In contrast to other Numeniini species[45,61], Eurasian Curlews are highly site-faithful, and only four of our studied curlews used more than one wintering site (i.e., moving further south after staying >20 d at the first wintering site).

Subsequently, the repeatability of the following parameters in consecutive years were analysed:

(1). *Latitude and longitude of wintering sites:* to determine the location of the wintering site for each bird in each year of observation, we computed the mean position of all GPS fixes recorded after arrival and prior to departure from the wintering ground (see below).

(2). *Latitude and longitude of breeding sites:* we determined the geographical location of the nest for each bird in each year of observation according to the method described by Bocher et al.[8], i.e. using a grid of 0.0001° x 0.0001° (corresponding to about 11.1 x 11.1 m) across the breeding site and defining the centre of the cell with the most GPS fixes as the position of the nest site. If the nest position was unknown, we used the mean position of all GPS fixes recorded after arrival in the breeding ground and prior to departure (similar to *(1)* and according to 20 km buffer in *(3)* below).

(3). *Arrival at and departure from wintering and breeding sites:* we assessed the start (departure) and end (arrival) dates of each migration by determining the time that the bird entered (or left) a 20 km buffer around its wintering or breeding site, respectively, as described by Pederson et al.[21]. The spatial buffer was created to avoid counting pauses close to the wintering and breeding sites as stopovers, given that birds may be vagrant shortly before and after departure/arrival in their breeding/wintering grounds.

(4). *Duration of spring and autumn migration:* the start and end dates (i.e. departure and arrival dates – see *(3)* above) were used to define the duration of spring and autumn migration, respectively.

(5). *Linear distance between breeding and wintering sites:* linear distance was defined as the Haversine (great circle) distance between positions of departure and arrival (see also *(3)* above).

(6). *Flown distance between breeding/wintering sites:* flown distance was measured by converting all GPS fixes of a given track into a line in QGIS[62]. The flown distance was the overall length of the resulting line.

(7). *Total stopover length:* stopovers were identified using flight speeds, which were computed by the temporal and spatial differences between consecutive tracking points. A threshold of 24 km/h was used to distinguish between flight and stopover behaviour, in accordance with Pederson et al.[21], and based on the analysis of bimodal speed histograms calculated from tracking data. The overall time of all stopover positions per individual was used to compute the total stopover length.

(8). *Mean stopover length:* the duration spent at a single stopover site was used to compute mean stopover length (as described in *(7)*).

(9). *Total stopover number:* a stopover was defined as a period of ≥ 60 min of stopover behaviour at a single spot, in accordance with Pederson et al.[21], to avoid including vagrant and flocking behaviours during migration. According stopovers (as defined in *(7)*) were counted per individual to give the total stopover number.

## Statistics and reproducibility

Previous studies showed that migration parameters, such as departure and arrival dates from/at the breeding and wintering sites, as well as linear and flown distances and stopover numbers and lengths, were significantly dependent on the location of the wintering and breeding sites[21,26]. This is mainly because curlews show chain migration, with birds wintering further south breeding further south and those wintering further north breeding further north. Birds breeding at higher latitudes thus need to wait for the snow to melt before they depart for their migration and/or need to adjust their stopovers with respect to green-up date[36]. The current study aimed to compare the repeatability of migration parameters within and between individuals, and it was therefore necessary to compare individuals from similar breeding and wintering clusters to avoid spatial and temporal biases in the inter-individual comparisons of migration routes and phenology. To

obtain information on the repeatability of migration patterns for a given individual compared with its conspecifics from similar breeding and wintering sites, we pooled data for birds that both bred and wintered within the same 250 km clusters to compare migration parameters during autumn and spring passage, separately (see Supplementary Fig. S3 for location of clusters and number of birds).

To assess repeatability, we computed the repeatability index $R$ using the R-package *rptR*[24,63]. Following this method, the repeatability measure was calculated as: $R = \sigma^2_{between}/\sigma^2_{total}$ where $\sigma^2_{between}$ represents the variance among individuals, and $\sigma^2_{total}$ is the total variance (including within-individual variation and measurement error). $R$ quantifies the proportion of total variance attributable to differences between individuals. $R = 0$ indicates no repeatability effect, meaning that all variation is caused by within-individual fluctuations or measurement error. Conversely, if $R = 1$, all variance is caused by between-individual differences, meaning that each individual consistently produces the same values across repeated measurements. We estimated repeatability (R, intraclass correlation) using *rptR*, with individual ID as a random intercept. To account for spatial grouping, we fitted cluster as a fixed effect and report adjusted repeatabilities[64], thereby controlling for systematic differences among breeding and wintering regions. This specification allows us to separate consistent individual differences from spatially structured variation and avoids ad-hoc data transformations or exclusion of small clusters. Sensitivity cheques using alternative model specifications yielded qualitatively similar results; we therefore base our inference on the fixed-effect formulation. Continuous traits were analysed with Gaussian linear mixed effect models[64] of the form y~cluster+log(dt+1) + (1|ID). For the count variable total stopover number, we used Poisson generalised linear mixed effect models[64] with an overdispersion check; if overdispersion was substantial we applied a squareroot-transformation and fitted a Gaussian model as fallback. We obtained 95% bootstrap confidence intervals (1,000 iterations) and permutation *p*-values as implemented in rptR. We finally visualised the repeatability by plotting the parameter of interest in the first observation year for each individual curlew against the observation for the consecutive year, as described by Kürten et al.[16]. Confidence intervals and *p*-values were created based on bootstrapping using $n = 1,000$ resamples. As an additional validation, $R$ was calculated for randomly re-ordered values independent of individuals (this showed $R \approx 0$, as expected). To overcome possible problems of different fix intervals between GPS tags due to differences in battery charge (see Pederson et al.[21]), we finally integrated the possible influence of average step size (dt) between fixes when calculating $R$ by integrating the logarithm of the time between consecutive GPS fixes as a fixed-effect predictor, thus correcting for the (possible) effect of different fix intervals. Our baseline models included cluster ID and log_dt as fixed effects and individual ID as the random effect of interest (e.g. y ~ cluster + log_dt + (1 | ID)). To account for possible group-level effects, we also tested sensitivity models including sex and capture country as additional fixed effects. Furthermore, as long-distance migrants were found to exhibit less variation in the timing of their migration compared with short-distance migrants[27,28], we conducted a post-hoc analysis to assess whether repeatability in migration-related traits depends on migration distance. Individuals were assigned to three distance classes (i.e. short, medium and long-distance migrants) based on the distribution of linear migration distances. For each season, repeatability estimates were rescaled within each trait to focus on relative differences among distance classes. We tested for a monotonic trend across classes using a regression of rescaled repeatability on distance class. Results are shown in Supplementary Fig. S2.

All statistical analyses were performed in R[65], version 4.4.2.

## Assessment of track similarity

The similarity of the location of migratory tracks was assessed as described by Guilford et al.[23] for Atlantic puffins (*Fratercula arctica*). We computed the nearest distance of each GPS fix of the first year's track to the nearest neighbour of the next year's track for the same individual (separately for autumn and spring migration; see Supplementary Fig. S4) using the

extension *nngeo::st_nn* in R[66]. Two measures for within-individual track similarity were then achieved by calculating the overall mean and overall median of all nearest-neighbour distances of these consecutive tracks. This was repeated for all tracks for the same individual. We then computed between-individual track similarity for all birds located in the same 250 km breeding and wintering clusters (Supplementary Fig. S3, see above) in the same way (i.e. computing the nearest distance of each GPS fix between all the different routes of birds within the same clusters, separately for each migration season). We then compared within-individual track similarity and between-individual track similarity using a generalised linear mixed model with a negative binomial distribution[67]. The mean and median distance between the tracks were then used as dependent variables, the cluster ID and bird ID were used as random intercepts, and the two factors migration season (with the levels "spring" and "autumn") and track type (with the levels "same individual" and "different individuals") were used as predictors. We computed an overall mean and 95% confidence intervals for the mean and median distance of within- and between-individual track similarity.

## Ethics

Tagging, catching and handling of curlews complied with European laws. In Germany, the procedure was approved by the Ministerium für Landwirtschaft, ländliche Räume, Europa und Verbraucherschutz of the federal state of Schleswig-Holstein (file numbers V 312-7224.121-37(42-3/13), V 241-35852/2017(88-7/17) and V 242-39334/2022(41-5/22)) and by the Lower Saxony State Office for Consumer Protection and Food Safety (LAVES) (file number 33-19-42502-04-17/2699 and 33-19-42502-04-22-00200). Tagging of curlews in NW Germany was approved by LAVES (file numbers 33.19-42502-04-20/3373 and 33.19-42502-04-21/3614), Senator for Labour, Women's Affairs, Health, Youth and Social Affairs in Bremen (33.19-42502-04-20/3373) and the State Agency for Nature, Environment and Consumer Protection North Rhine-Westphalia (LANUV, file numbesr81.02.04.2020.A097 and 81-02.04.2021.A128). Capture and tagging of curlews on the German side of the Upper Rhine Valley was approved by Regierungspräsidium Freiburg (G-17/53). The GPS-tagging of curlews in Bavaria was approved by the regional government of Lower Franconia Veterinary Office and Consumer Protection (file numbers RUF 55.2.2-2532.2-247 and 55.2.2-2532.2-1355), and by the Higher Nature Conservation Agencies of the regional governments of Middle Franconia, Lower Bavaria, Upper Bavaria and Upper Palatinate. Permission to tag curlews in France was granted by the Centre de Recherches sur la Biologie des Population d'Oiseaux (file numbers PP336 and PP1083). Tagging in Estonia was approved by the Matsalu Ringing Centrer, Estonian Environmental Agency (file number 3-2013 and 4-2013 within the "Programme of marking Eurasian curlew"). Treatment of curlews in Finland was done according to the permissions issued by the Centre for Economic Development, Transport and the Environment (file number VARELY/1136/2020 and VARELY/3622/2017). Tagging in Poland was approved by the General Directorate for Environmental Protection (file numbers DZP-WG.6401.03.98.2016.km, DZP-WG.6401.03.97.2017.jro, DZP-WG.6401.03.2.2018.jro, DZP-WG.6401.102.2020.TŁ).

## Reporting summary

Further information on research design is available in the Nature Portfolio Reporting Summary linked to this article.

## Data availability

All numerical source data for graphs and charts can be found in Supplementary Data 1-5. Raw data of tagged curlews are stored on www.movebank.org and can be made available upon request by the corresponding author.

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

## Acknowledgements

Many field workers assisted with bird catching and handling. In particular, we would like to thank G (Niko) Nikolaus, the Schutzstation Wattenmeer e.V., T Sveridova, R Kylmänen, T Seimola, P Delaporte, R Vohwinkel, J Schmidt, G Müskens and the team from Nature Reserve of Moëze-Oléron. We thank Sue Furness for editing a draft of the manuscript. Finally, we thank Nils Warnock and Ben Lagasse who reviewed the manuscript and made very valuable comments.

## Author contributions

P.S. collected data, supervised the research, analysed the data, wrote the original draft; M.D. wrote the manuscript, collected and analysed the data; M.M. performed the statistical analyses, S.G., M.B., H.D., J.E., T.F., W.F., F.J., S.K., M.K., H.K., D.K., R.M., M.P., P.R., V.R., and P.B. collected the data and revised the manuscript. All authors contributed to editing and approving the submitted version.

## Funding

Parts of the study were conducted within the project "Triple crises meets trilateral cooperation: effects of biodiversity loss, climate change and pollution on salt marshes & pathways to their sustainable management" Tricma² funded by the German Federal Ministry of Research, Technology and Space (BMFTR) and the Federal Ministry for the Environment, Nature Conservation and Nuclear Safety (BMUKN) under the grant number (FKZ 03F0960C). Furthermore, parts of the study were funded by the German Federal Agency for Nature Conservation (Bundesamt für Naturschutz, BfN) within the project "Trackbird" (FKZ 3519861400) and "OWP Vogelzug" (FKZ 352315100 A) with funds from the German Federal Ministry for the Environment, Climate Action, Nature Conservation and Nuclear Safety (BMUKN). Tagging in NW Germany was funded by the BfN (FKZ 3520 53 2052) with funds from the BMUKN and the Lower Saxony Water Management, Coastal Defence and Nature Conservation Agency (NLWKN) with funds from the Lower Saxonian Ministry for the Environment Energy and Climate Protection, Niedersächsische Bingo Umweltstiftung, the Senator for Climate Protection, Environment, Mobility, Urban Development and Housing of the State of Bremen, the National Park Administration of the Wadden Sea National Park of Lower Saxony, Niedersächsische Wattenmeer-Stiftung, NRW-Stiftung, Naturschutzstiftung des Landkreises Osnabrück, Naturschutzstiftung Kreis Steinfurt, the districts of Aurich, Cloppenburg, Cuxhaven, Diepholz, Emsland, Grafschaft Bentheim, Leer and Verden. Tagging of birds in south Germany were funded by Bayerischer Naturschutzfond and Bayerisches Landesamt für Umwelt. Work in Estonia was funded by the Estonian Environmental Investment Centre and by the project "Applied research for conservation of the Eurasian curlew" funded by the Estonian Environmental Investment Centre (grant No. 8172). Research in France was funded by the French Ministry of Ecology (project BirdMan), Agence Française de Développement, (project Limitrack, grant QUALIDRIS), Contrat de Plan Etat-Région, CNRS (grant ECONAT), Ligue pour la Protection des Oiseaux. For PB and FJ, this study was also funded by the European Union through its HORIZON Research and Innovation Actions (HABITRACK, Project 101135047, HORIZON-CL6-2023-BIODIV-01 Biodiversity and Ecosystem Services). Funding in Poland was provided by the EU Cohesion Fund under the Operational Programme Infrastructure and Environment 2014-2020, under the project POIS.02.04.00-00-0019/16 entitled "Implementation of the National Action Plan for Eurasian Curlew - stage I", coordinated by the Wildlife Society "Stork". Open Access funding enabled and organized by Projekt DEAL.

## Competing interests

The authors declare no competing interests.

## Additional information

[1]Research and Technology Centre (FTZ), University of Kiel, Kiel, Germany. [2]Littoral Environnement et Sociétés (LIENSs), UMR 7266 La Rochelle University - CNRS, La Rochelle, France. [3]Bionum GmbH – Consultants in Biological Statistics, Hamburg, Germany. [4]Bioplan Bühl, Bühl, Germany. [5]Niedersächsisches Ministerium für Umwelt, Energie und Klimaschutz, Hannover, Germany. [6]BirdLife Estonia, Tartu, Estonia. [7]Department of Biodiversity and Landscape Ecology, Osnabrück University, Osnabrück, Germany. [8]Institute of Biodiversity and Landscape Ecology (IBL), Münster, Germany. [9]Max Planck Institute of Animal Behavior, Radolfzell, Germany. [10]Centre for the Advanced Study of Collective Behaviour, University of Konstanz, Konstanz, Germany. [11]UMR7204 CESCO, Museum National D'Histoire Naturelle, CNRS, Sorbonne Université, Paris, France. [12]Museum & Institute of Zoology, Polish Academy of Sciences, Warszawa, Poland. [13]Nature Association Dubelt, Michałowo, Poland. [14]Institute for Wetlands and Waterbird Research e.V., Verden, Germany. [15]Towarzystwo Przyrodnicze "Bocian" (Wildlife Society "STORK"), ul, Warszawa, Poland. [16]"Lendület" Landscape and Conservation Ecology, Institute of Ecology and Botany, HUN-REN Centre for Ecological Research, Vácrátót, Hungary. [17]Natural Resources Institute Finland, Helsinki, Finland. [18]Finnish Museum of Natural History, P. Rautatiekatu 13, 00101 University of Helsinki, Helsinki, Finland. [19]National Nature Reserve of Moëze-Oléron, LPO Ligue pour la Protection des Oiseaux, Plaisance, France. [20]Landesbund für Vogel- und Naturschutz in Bayern e.V., Hilpoltstein, Germany. ✉e-mail: schwemmer@ftz-west.uni-kiel.de

