## [Transparent Peer Review file · Communications Biology]

High annual-cycle repeatability suggests low flexibility to environmental changes in a near-threatened migratory shorebird

Corresponding Author: Dr Philipp Schwemmer

Version 0:

Reviewer comments:

Reviewer #1

(Remarks to the Author)

Hi,
This is an ambitious paper and the data that were collected are difficult to obtain. I applaud you all for your effort. I have some general comments/questions for you to consider:
Methodologically, it is still unclear to me how you defined the non-breeding areas (and I suggest using the term non-breeding instead of wintering) for the curlews? From work on various of the Numeniini, they tend to wander on their non-breeding grounds and for us it was always interesting trying to define where their non-breeding grounds began and ended within the annual cycle. Do Eurasian Curlews move on their non-breeding grounds? As you indicated, there is a fair amount of literature on them moving in response to weather variables like cold weather. Some discussion of how you determined where the curlew's non-breeding grounds began and ended would be appreciated. Might look at Page et al. (2014), Ruthrauff et al. (2021) for some other ways people have defined this.
Analytically, you explained why you did not use age of curlews in the analysis, and I was wondering why variables like sex and capture location are not included in your analyses since you acknowledge their effect on the migratory behavior of these birds. Seems like they might account for a fair amount of variation in your analyses and there might be interesting differences, especially by sex.
Some discussion of why you chose to use 250km radius clusters of birds would be informative. You have so many clusters that it tends to overshadow your story. Do your results change much if you use say a 500km radius? The influence of spatial scale is not discussed, and this has lots of ramifications for repeatability measures.
You discuss the number of stopover sites used but then you say (l. 310) that you do not know if curlews use the same stopovers repeatedly. That seems odd to me since you seem to have the data to identify stopovers, and in the beginning of the manuscript you say that the Wadden Sea was consistently used and was a very important site. Some explanation of why you do not present these data would be helpful.
I thought that your paper could benefit from some broader comparisons (examples from birds > Numeniini > curlews). Currently, it mostly focuses on Eurasian Curlews and European studies. Our knowledge on the movement ecology of birds, including shorebirds, and the Numeniini, is exploding and there are many relevant studies out there. In particular, there are some relevant studies of Bar-tailed Godwits (e.g. Battley 2006, Conklin et al. 2013) worth discussing. Whimbrel and other shorebirds have been extensively tracked and studied with discussions about repeatability of different aspects of the migratory behavior (Carneiro et al. 2019, Ruthrauff et al. 2019, 2021
I would suggest also discussing some of the limitations of repeatability measures (e.g. Stuber et al. 2022, follow up comments by Dingemanse et al. 2022). The discussion by Conklin et al. (2013) is especially relevant.
Finally, since you discuss the potential that the strong repeatability in various aspects of the curlew's annual cycle is maladaptive, it would be worth a bit of discussion of what the consequences of the maladaptive behavior would look like and in response to what threats. Are there specific threats for Eurasian Curlews that high repeatability behavior would make them more vulnerable to? What does their threat landscape look like? In Long-billed Curlews, one threat is the loss and conversion of agricultural habitat. For certain races of the Bar-tailed Godwit, it is the loss of tidal habitat in the Yellow Sea as shown by Piersma et al. (2016). As a side note, it would be very interesting to link your repeatability data with survival data broken down by seasons.
Some specific questions that reference specific lines:
l. 92 – you say that you need high resolution data for repeatability studies, yet the Franklin et al. (2022) review suggests that

this is not the case. Maybe rephrase.

I. 119 – missing something after “... against the background of current and future needs [of what?] to adjust to...”

I. 129 – how did you conclude that the Wadden Sea was the most important stopover site? Largest site? Most used? Explain.

I. 137 – sometimes you use “sites” and other times you use “areas”. How are you defining these? Be consistent in use.

I. 141 - Fig. 2 - Figure 2 is difficult to read and to see where the breeding and non-breeding locations are. Perhaps make the symbols for breeding and non-breeding location larger and strengthen width of lines. Would it be possible to show these on the same background map (instead of in four maps)?

I. 162 – Fig 3 – I did not find this figure particularly informative. Could this be dropped and the results summarized in the text? Other ways to graph these data? i.e. Vardanis et al. (2011)

I. 181 (Table 1) – Suggest adding sample sizes to the table

Lines 186-193 – P values should be accompanied by test scores and df, unless you can find that information elsewhere. Do not see that in Fig. 6 that is referenced.

L 209-211 – This is a large study with multiple years of data on the same individuals. However, I would soften this statement some. The terms “large” and “high” are subjective, and there are some good studies out there with solid sample sizes and also have multiple years of data (e.g. Conklin et al. 2013, Léandri-Breton et al. 2021, Kürten et al. 2022). Maybe say that studies like yours are uncommon. If you want to say this study is the first for large numbers and years, I suggest including a table with some of the major studies (or at least reference the currently most significant ones).

L. 220-222 – but this does not necessarily mean that they are not flexible in response to threats. Do you have any specific examples?

I. 235-237 – Seems like you should have the data to test this?

I. 238 – what do other studies say about this? This would be a good place to provide some comparison with other studies (examples from birds > Numeniini > curlews). Perhaps put your study in the context of the Franklin et al.’s (2022) review of repeatability and migration phenology?

I. 248 – “...the reason for this behavior is unknown.” – What do other studies suggest?

I. 290-295 – Here you say that the curlews show low flexibility to react to environmental change but this seems to be at odds with your discussion above (roughly I. 274-280) where you say that they do not follow the same migration routes and use different stopover sites (and in I. 302 where you say that they are flexible). In the abstract (I. 66) you call curlews a “... stereotypical near threatened species.” Your messaging about whether you think that Eurasian Curlews are flexible or inflexible is inconsistent.

L. 303-305 – Conklin et al. (2021) show that individual Bar-tailed Godwits are able to adjust migration strategies

L. 308-309 – repeats what you say in I. 214-217.

Some of the literature I cited (or looked at for this review):

Battley, P.F., 2006. Consistent annual schedules in a migratory shorebird. *Biology Letters*, 2(4), pp.517-520.

Carneiro, C., Gunnarsson, T.G. and Alves, J.A., 2019. Why are whimbrels not advancing their arrival dates into Iceland? Exploring seasonal and sex-specific variation in consistency of individual timing during the annual cycle. *Frontiers in Ecology and Evolution*, 7, p.248.

Conklin, J.R., Battley, P.F. and Potter, M.A., 2013. Absolute consistency: individual versus population variation in annual-cycle schedules of a long-distance migrant bird. *PLoS One*, 8(1), p.e54535.

Conklin, J.R., Lisovski, S. and Battley, P.F., 2021. Advancement in long-distance bird migration through individual plasticity in departure. *Nature Communications*, 12(1), p.4780.

Dingemanse, N.J., Hertel, A.G. and Royauté, R., 2022. Moving away from repeatability: a comment on Stuber et al. *Behavioral Ecology*, 33(3), pp.488-489.

Franklin, K.A., Nicoll, M.A., Butler, S.J., Norris, K., Ratcliffe, N., Nakagawa, S. and Gill, J.A., 2022. Individual repeatability of avian migration phenology: A systematic review and meta-analysis. *Journal of Animal Ecology*, 91(7), pp.1416-1430.

Léandri-Breton, D.J., Tarroux, A., Elliott, K.H., Legagneux, P., Angelier, F., Blévin, P., Bråthen, V.S., Fauchald, P., Goutte, A., Jouanneau, W. and Tartu, S., 2021. Long-term tracking of an Arctic-breeding seabird indicates high fidelity to pelagic wintering areas. *Marine Ecology Progress Series*, 676, pp.205-218.

Page, G.W., Warnock, N., Tibbitts, T.L., Jorgensen, D., Hartman, C.A. and Stenzel, L.E., 2014. Annual migratory patterns of Long-billed Curlews in the American West. *The Condor: Ornithological Applications*, 116(1), pp.50-61.

Piersma, T., Lok, T., Chen, Y., Hassell, C.J., Yang, H.Y., Boyle, A., Slaymaker, M., Chan, Y.C., Melville, D.S., Zhang, Z.W. and Ma, Z., 2016. Simultaneous declines in summer survival of three shorebird species signals a flyway at risk. *Journal of Applied Ecology*, 53(2), pp.479-490.

Ruthrauff, D.R., Tibbitts, T.L. and Gill Jr, R.E., 2019. Flexible timing of annual movements across consistently used sites by Marbled Godwits breeding in Alaska. *The Auk: Ornithological Advances*, 136(1), p.uky007.

Ruthrauff, D.R., Harwood, C.M., Tibbitts, T.L., Warnock, N. and Gill Jr, R.E., 2021. Diverse patterns of migratory timing, site use, and site fidelity by Alaska-breeding Whimbrels. *Journal of Field Ornithology*, 92(2), pp.156-172.

Stanley, C.Q., MacPherson, M., Fraser, K.C., McKinnon, E.A. and Stutchbury, B.J., 2012. Repeat tracking of individual songbirds reveals consistent migration timing but flexibility in route. *PloS one*, 7(7), p.e40688.

Stuber, E.F., Carlson, B.S. and Jesmer, B.R., 2022. Spatial personalities: a meta-analysis of consistent individual differences in spatial behavior. *Behavioral Ecology*, 33(3), pp.477-486.

van Wijk, R.E., Bauer, S. and Schaub, M., 2016. Repeatability of individual migration routes, wintering sites, and timing in a long-distance migrant bird. *Ecology and evolution*, 6(24), pp.8679-8685.

Vardanis, Y., Nilsson, J.Å., Klaassen, R.H., Strandberg, R. and Alerstam, T., 2016. Consistency in long-distance bird migration: contrasting patterns in time and space for two raptors. *Animal Behaviour*, 113, pp.177-187.

I hope these comments are useful. Feel free to contact me if you have any questions.

Reviewer #2

(Remarks to the Author)

This manuscript examines within-individual repeatability of spatial and temporal migration patterns of Eurasian Curlews using an impressive collection of GPS tracks. The results indicate that Eurasian Curlews exhibited high repeatability in spatial and temporal migration patterns. The authors conclude that Eurasian Curlews potentially lack the flexibility required to adaptively respond to environmental change.

In general, I liked the manuscript. The length is great, the figures are nice, and the authors have taken care in putting the manuscript together. I also feel, however, that (1) there are important aspects of the repeatability models that need to be addressed, and (2) there is very little support for the conclusion that the lack of flexibility exhibited by the tracked curlews represents a potentially maladaptive response to environmental change or ecological trap. For example:

#1: For a bird like a Eurasian Curlew, which has high annual survival and a relatively slow life history strategy, high repeatability and prioritizing safety can be a highly adaptive response to environmental change — particularly when facing low predictability and high variability in environmental conditions. For discussions of tradeoffs associated with high repeatability versus flexibility under varying environmental conditions, see:

Lof et al. 2012. Timing in a fluctuating environment: Environmental variability and asymmetric fitness curves can lead to adaptively mismatched avian reproduction. <https://doi.org/10.1098/rspb.2012.0431>

Bauer et al. 2020. Environmental variability, reliability of information and the timing of migration. <https://doi.org/10.1098/rspb.2020.0622>

#2: The manuscript doesn't offer any insight on the extent to which the areas in which Eurasian Curlews bred, migrated, and wintered underwent environmental change. Could the Eurasian Curlews in the study have been capable of surprising flexibility (e.g., lines 261–262) but exhibited high repeatability because of the relative stability of the environments in which they occurred, during the time periods in which they were in them? To conclude that the Eurasian Curlews in the study were potentially maladapted, I think (at a minimum) it would need to be shown that the curlews in the study exhibited high repeatability despite significant environmental changes in the locations and time periods in which they were tracked.

#3: Environmental changes don't necessarily require that individuals alter their migrations. For example, environmental changes during individuals' lifespans may not alter resource distributions to the point that individuals need to flexibly adjust their migrations. Also, individuals with high repeatability in migratory timing may adjust to environmental changes by flexibly adjusting other aspects of their annual routines (Carneiro et al. 2019).

Carneiro et al. 2019. Why are Whimbrels not advancing their arrival dates into Iceland? Exploring seasonal and sex-specific variation in consistency of individual timing during the annual cycle. <https://doi.org/10.3389/fevo.2019.00248>

Some of the above considerations are mentioned in the Discussion but may warrant greater emphasis. I would also consider (1) reframing the manuscript to explore the extent to which Eurasian Curlews might adapt to environmental changes via processes of phenotypic selection versus phenotypic flexibility, and (2) focusing the take home message of the manuscript on important next steps, conservation implications, or something of the sort.

Additional comments can be found below.

All the best,
Ben Lagasse

Comment #1; Lines 52–66:

Consider (1) reframing the manuscript to explore the extent to which Eurasian Curlews might adapt to environmental changes via processes of phenotypic selection versus phenotypic flexibility, and (2) focusing the take home message of the manuscript on important next steps, conservation implications, or something of the sort.

Comment #2; Line 248:

I'm thinking Eurasian Curlews molt on their wintering grounds. If so, it could be worth mentioning that here and providing a brief thought on how consistent arrival timing on their wintering grounds might (or might not) be tied to individuals' growing high-quality flight feathers in an optimal way. For example, see:

McNamara et al. 1998. The timing of migration within the context of an annual routine. <https://doi.org/10.2307/3677160>

Barta et al. 2008. Optimal moult strategies in migratory birds. <https://doi.org/10.1098/rstb.2007.2136>

Comment #3; Lines 308–322:

The prior paragraph (lines 290–307) concludes the manuscript. I would remove this paragraph or place it before lines 290–307.

Comment #4; Lines 427–438:

There are multiple aspects of the modeling approach that could be good to address:

First, I'm thinking that cluster ID should be fitted as a fixed effect, not a random effect. Unlike modeling exercises in which the fixed effects are the parameters of interest, in repeatability models the random effects are the parameters of interest (Stoffel et al. 2017, page 3). Also, with cluster ID fitted as a fixed effect, mean-centering the data by cluster ID isn't necessary, as there would be a beta estimate per cluster that accounts for differences in means.

Correctly specifying the model is probably preferable to centering the data by cluster ID (to deal with the non-convergence issue) because it incorporates the hierarchical structure of the data (individuals within clusters) in the model.

Second, it is somewhat unclear if the manuscript reports "agreement repeatabilities" or "adjusted repeatabilities" or "enhanced agreement repeatabilities" (Stoffel et al. 2017). As stated, the modeling approach is designed to account for the effect of different GPS fix intervals and cluster IDs, and thus I'm thinking the intention is to report adjusted repeatabilities. Consider clarifying that the manuscript reports adjusted repeatabilities.

Third, alongside fitting cluster ID as a fixed effect, it could be worth revisiting the use of Gaussian versus Poisson models. For example, consider using the distribution that describes each migration parameter rather than a one-size-fits-all approach. Poisson distributions are intended to describe counts over time and thus should probably be used to estimate repeatability in total stopover number.

Version 1:

Reviewer comments:

Reviewer #1

(Remarks to the Author)

Hi,

Thanks to your careful attention to comments in my first review of your interesting paper and your detailed answers.

I have attached a word file with relatively minor edits and a few questions/comments.

Feel free to contact me with any further questions.

Best,

Nils

Nils Warnock (nwarnock@allhandsecology.org)

Reviewer #2

(Remarks to the Author)

Many thanks to the author(s) for addressing my comments. Below are additional thoughts. I hope they help. All the best,

Line 57: Consider replacing "entire annual cycle" with "entire migratory cycle." The term, entire annual cycle, could include examining nest initiation dates, initiation of molt, completion of molt, etcetera, which were not addressed. Entire migratory cycle seems more precise.

Lines 167–175: It's not clear to me why this comparison was needed. It seems that treating wintering and breeding cluster as a fixed effect is the correct specification of the model and there is no need to compare the correctly specified models against mean-centered (i.e., incorrectly specified) models. Consider removing the comparison from the manuscript or elaborating in the Methods (i.e., Lines 534–536) why this comparison was needed.

Figure 3–5, and Table 1: Consider moving Figure 3–5 and Table 1 to the appendix and replacing them with a single figure that plots the corresponding R values and their 95% intervals. Such a figure would need to include corresponding samples sizes, and could distinguish between "significant" and "nonsignificant" results. Doing so would streamline the manuscript without a significant loss of information and would allow rapid and intuitive comparisons of repeatability values between migration parameters. For example, a single figure with R values and their intervals could plainly illustrate 1) the high repeatability of breeding and wintering sites, 2) the higher repeatability of spring migration dates compared to autumn

migration dates, and 3) higher stopover repeatability in spring compared to autumn.

Lines 266–269: I'm not seeing any mention of this analysis in the Methods or Results. It would be good to clarify that this was a post hoc analysis and present the results in a supplemental figure. Alternatively, the analysis could be included in the Methods and Results.

Lines 299–301: Consider rephrasing this sentence along the lines of, "Although short-term cold spells during the years of our study may have decreased in frequency or severity compared to decades ago (i.e., reference #37), curlews have been described to...."

Only observing such behavior in 2 individuals may reflect a lack of flexibility as much as it might reflect a decrease in the frequency and severity of cold spells.

Lines 302–303: I don't think the results support this statement. At a minimum, it would be good to revise the phrase, "...are less able to..." This manuscript doesn't really examine what curlews are or aren't able to do. It describes what curlews happened to do during the years of the study. For example, I think Lines 355–357 do a nice job of describing the results — i.e., "suggests a low flexibility" and "despite this apparent lack of flexibility".

Lines 304: Again, I think it would be good to revise the phrase, "...be able to..."

Lines 309: "Environmental plasticity" doesn't seem quite right here. Consider clarifying that the population as a whole might adapt.

Line 352–355: Consider replacing "entire annual cycle" with "entire migratory cycle." The term, entire annual cycle, could include examining nest initiation dates, initiation of molt, completion of molt, etcetera, which were not addressed. Entire migratory cycle seems more precise.

Line 399: "phenological plasticity" doesn't seem quite right here. Consider clarifying that, "...if curlews lack the flexibility to adapt..."

Thanks a lot for giving us the opportunity to revise our manuscript entitled “High annual-cycle repeatability suggests low flexibility to environmental changes in a near-threatened migratory shorebird” along the very valuable and constructive suggestions made by the two reviewers and yourself.

Please find on the following pages our response (**in bold**) to the questions raised by the two reviewers (*in italics*). Line numbers mentioned by the reviewers refer to the original manuscript version, line numbers stated in our answers refer to the revised version.

Comments by reviewer #1 :

1. *This is an ambitious paper and the data that were collected are difficult to obtain. I applaud you all for your effort.*

Thanks a lot for acknowledging the value of our data and your positive feedback.

2. *Methodologically, it is still unclear to me how you defined the non-breeding areas (and I suggest using the term non-breeding instead of wintering) for the curlews? From work on various of the Numeniini, they tend to wander on their non-breeding grounds and for us it was always interesting trying to define where their non-breeding grounds began and ended within the annual cycle. Do Eurasian Curlews move on their non-breeding grounds? As you indicated, there is a fair amount of literature on them moving in response to weather variables like cold weather. Some discussion of how you determined where the curlew’s non-breeding grounds began and ended would be appreciated. Might look at Page et al. (2014), Ruthrauff et al. (2021) for some other ways people have defined this.*

Thank you for highlighting possible shortcomings in the definition of wintering sites. Indeed, we followed a very similar approach as stated in Page et al. (2014). Our definition of wintering sites is described in more detail in Donnez et al. (2023) Wetlands 43: 80 [<https://doi.org/10.1007/s13157-023-01728-w>]. Thus, we have now cited both papers in the methods section and additionally added a brief description of how we defined wintering sites and how we separated them from southernmost stop-over sites (ll. 446).

In contrast to many other Numeniini species, Eurasian curlews show an extremely high site-fidelity and do only rarely change wintering / non-breeding grounds – only four cases (4.3%) in our entire dataset (now also stated in the

text). Of these four, two curlews once moved further south to avoid cold temperatures (we have cited Düttmann et al. 2025). However, these movements during winter were exceptional and only appeared during cold temperatures. Please see also our comment no. 14 concerning Figure 3: this figure illustrates the surprisingly stationary behaviour of curlews during their wintering grounds which is constant during every year (this is also why we suggest to keep this figure). The site use itself is very restricted, i.e. mean homeranges across different individuals: 533 ha (95% Kernel densities) and mean core areas: 62 ha (50% Kernel densities) (Donnez et al. 2023). We have submitted another manuscript showing the extraordinarily high repeatability in small-scale space use within wintering sites (based on remote sensing data) which will complement the current study (and illustrate in detail the findings presented in Figure 3). We have now also referenced this study as “Bocher submitted” in the methods and discussion section.

With respect to the term wintering area vs. non-breeding area, we feel that the term wintering area is more appropriate because non-breeding areas would also comprise stop-over sites. Therefore, we suggest to stick with this term. Should the editor or reviewer feel that the term non-breeding area is more appropriate, we are open to change it.

- 3. Analytically, you explained why you did not use age of curlews in the analysis, and I was wondering why variables like sex and capture location are not included in your analyses since you acknowledge their effect on the migratory behavior of these birds. Seems like they might account for a fair amount of variation in your analyses and there might be interesting differences, especially by sex.*

Thanks a lot for highlighting this important point. In our revised analyses we now fitted cluster ID as a fixed effect (together with log_dt) and individual ID as the random effect of interest. To test whether sex (female/male) and capture location might explain additional variance, we have now extended these baseline models by including sex and catching country as fixed effects as suggested.

Across all migration parameters and both seasons, the adjusted repeatabilities from these sensitivity models were virtually identical to the baseline values ($\Delta R < 0.02$ in all cases). For example, for spring departure date the baseline model yielded $R = 0.537$ (95% CI: 0.209–0.728), and the adjusted model $R = 0.499$ (95% CI: 0.130–0.715). Neither sex nor capture location explained a significant proportion of variance. The only exception were weak tendencies in autumn

departure and arrival dates, where males departed and arrived slightly later than females (t-values ~ 2), but these effects were inconsistent and did not alter repeatability estimates. We added a corresponding paragraph in the results section (II. 179).

We have now clarified in the Methods that sex and capture country were tested as additional fixed effects in sensitivity analyses, and we mention in the Results that these covariates had no consistent effect and did not change our conclusions.

4. *Some discussion of why you chose to use 250km radius clusters of birds would be informative. You have so many clusters that it tends to overshadow your story. Do your results change much if you use say a 500km radius? The influence of spatial scale is not discussed, and this has lots of ramifications for repeatability measures.*

To test the robustness of our repeatability estimates against the spatial scale used to define breeding and wintering clusters, we now re-ran the repeatability models using alternative buffer radii of 150 km and 500 km in addition to the 250 km used in the main analysis.

Results were highly consistent between 150 km and 250 km buffers. Repeatability estimates were nearly identical across all migration parameters and both seasons (Pearson correlation = 0.98, mean $|\Delta R| = 0.035$, maximum $|\Delta R| = 0.105$; see Figure A below which we have now also included as Supplement 2). This demonstrates that the estimated individual consistency in migratory timing and space use is very robust to moderate changes in the clustering scale.

At a 500 km buffer, the number of spatial clusters decreased markedly. This over-aggregation reduced between-individual variance and led to numerical instability in several models (e.g breeding area longitude, linear and flown distance breeding / wintering site, respectively), some of which returned degenerate variance components with $R \approx 0$ (see Fig. B below which we have now also included as Supplement 2). Such singular fits are a known phenomenon in mixed-effects models when the estimated random-effect variance approaches zero or random effects become perfectly correlated (Bates et al. 2015, *J. Stat. Softw.*, 67:1–48, <https://doi.org/10.18637/jss.v067.i01>). We therefore interpret the 500 km results as unreliable and restrict inference to the stable 150 km and 250 km analyses.

Overall, these results confirm that the main conclusions of our study are robust to reasonable variation in the spatial buffer definition. We have added a corresponding paragraph to the manuscript (II. 183).

5. You discuss the number of stopover sites used but then you say (l. 310) that you do not know if curlews use the same stopovers repeatedly. That seems odd to me since you seem to have the data to identify stopovers, and in the beginning of the manuscript you say that the Wadden Sea was consistently used and was a very important site. Some explanation of why you do not present these data would be helpful.

Thanks a lot for this hint. We agree that it is important to point out in the manuscript why we did not have a closer look on the repeated use of stop-over sites. Indeed, this would require a complex additional analysis. This is why we are currently preparing separate analysis of the repeated use of stop-over sites in a follow-up manuscript. For a detailed study of the repeated / varying use of stop-over sites among years it would be necessary to include remote sensing data as terrestrial sites (often used by curlews as stop-over sites) might show a different structure of land cover. This would clearly exceed the scope of the current study. Therefore, we have now included several sentences in the discussion section explaining these issues. We had already mentioned the possibility of comparing the repeated use of stop-overs sites at the end of the discussion to provide an outlook for future studies. This is now clarified.

Concerning the use of the Wadden Sea, please see comment no. 11 below.

6. I thought that your paper could benefit from some broader comparisons (examples from birds > Numeniini > curlews). Currently, it mostly focuses on Eurasian Curlews and European studies. Our knowledge on the movement ecology of birds, including

shorebirds, and the Numeniini, is exploding and there are many relevant studies out there. In particular, there are some relevant studies of Bar-tailed Godwits (e.g. Battley 2006, Conklin et al. 2013) worth discussing. Whimbrel and other shorebirds have been extensively tracked and studied with discussions about repeatability of different aspects of the migratory behavior (Carneiro et al. 2019, Ruthrauff et al. 2019, 2021)

Thanks a lot for these additional papers. We have followed your suggestion and added several of the suggested studies as a comparison with our results. Given that we had already made several comparisons with other species (using some different papers) and given the current length of the revised manuscript, we did not use all of the suggested studies but added additional examples of bar-tailed godwits and whimbrels in the discussion section also to highlight cases from different regions (II. 326).

7. *I would suggest also discussing some of the limitations of repeatability measures (e.g. Stuber et al. 2022, follow up comments by Dingemanse et al. 2022). The discussion by Conklin et al. (2013) is especially relevant.*

Done. We have now discussed our findings using the suggested literature on repeatability measures with a special focus on the consistency of potential methodological error sources and also extended the discussion on intra- vs interindividual variances in repeatability measures. We now point out that our approach using mixed-effect models explicitly accounted for both, inter- and intra-individual variance and therefore we interpret repeatability as a relative measure of consistent individual differences, not as a measure of absolute consistency or evidence of stable personality traits (as done in the meta-analyses of Stuber et al. 2022 and Dingemanse et al. 2022). See II. 336

8. *Finally, since you discuss the potential that the strong repeatability in various aspects of the curlew's annual cycle is maladaptive, it would be worth a bit of discussion of what the consequences of the maladaptive behavior would look like and in response to what threats. Are there specific threats for Eurasian Curlews that high repeatability behavior would make them more vulnerable to? What does their threat landscape look like? In Long-billed Curlews, one threat is the loss and conversion of agricultural habitat. For certain races of the Bar-tailed Godwit, it is the loss of tidal habitat in the Yellow Sea as shown by Piersma et al. (2016). As a side note, it would be very interesting to link your repeatability data with survival data broken down by seasons.*

We agree that a description of the consequences of missing flexibility in curlews needs to be discussed in more detail and that we need to point out the concrete threats in the landscape for curlews with respect to their strong repeatability of their annual cycle. Therefore, we are now discussing that wintering sites in western Europe are comparatively well protected against anthropogenic impacts (in contrast to the loss of tidal flats in the Yellow Sea; Piersma et al. 2016). However, climate change induced changes such as sea level rise is able to impact roosting and foraging sites in the wintering grounds also in conservation areas such as the World Heritage Site of the Wadden Sea. However, these environmental changes act over longer time spans and cannot be solved within the time period covered by this study (II. 363). In contrast, there is evidence of risks of short(er)-term anthropogenic and weather effects on curlews which is clearly increased by strong repeatability / missing flexibility. We have now included two examples of this, i.e. ongoing afforestation in north-west Russian breeding sites (II. 249) as well as sudden cold spells within the wintering sites of curlews (II. 299).

Thank you also very much for pointing out the idea to connect repeatability data with survival data by seasons. Unfortunately, we are not able to assess survival rates of our tagged birds as the end of data transmission does not necessarily indicate mortality, because tags might just stop working or fall off the birds. Regular re-sightings of our focal animals are also not possible given the large-scale and inaccessible wintering habitats and remoteness of most breeding sites in Russia.

9. *I. 92 – you say that you need high resolution data for repeatability studies, yet the Franklin et al. (2022) review suggests that this is not the case. Maybe rephrase.*

We agree and have rephrased the sentence accordingly and avoided the term “at high resolution”.

10. *I. 119 – missing something after “...against the background of current and future needs [of what?] to adjust to...”*

We have rephrased the sentence.

11. I. 129 – *how did you conclude that the Wadden Sea was the most important stopover site? Largest site? Most used? Explain.*

The Wadden Sea was used by most of the curlews tagged in our international tagging dataset. This is now clarified in the text.

12. I. 137 – *sometimes you use “sites” and other times you use “areas”. How are you defining these? Be consistent in use.*

We agree and have now consistently used the term “sites”. In contrast to “areas”, this term mirrors better the spatially restricted behavior of the birds which is more in line with the data presented.

13. I. 141 - *Fig. 2 - Figure 2 is difficult to read and to see where the breeding and non-breeding locations are. Perhaps make the symbols for breeding and non-breeding location larger and strengthen width of lines. Would it be possible to show these on the same background map (instead of in four maps)?*

Thanks a lot for this hint. We have now altered the maps and inserted larger symbols and wider lines as suggested. We tested but rejected the idea of showing everything on the same background map as clarity was strongly reduced.

14. I. 162 – *Fig 3 – I did not find this figure particularly informative. Could this be dropped and the results summarized in the text? Other ways to graph these data? i.e. Vardanis et al. (2011)*

We actually feel that the figure is very well showing the extremely high site-faithfulness to wintering and breeding sites as the data highlight that the sites are absolutely the same during consecutive years (with only minor and variation – e.g. short-term impacts of cold spells in few individuals) and that they are also very small. Both results are a core finding in our study deserving an according illustration. We think that the strong consistency of wintering and breeding sites can be depicted nicely from the current graph. Should the editor or reviewer have a different opinion, we are happy to provide other solutions.

15. I. 181 (Table 1) – *Suggest adding sample sizes to the table*

Done. The table now reports the slightly changed values (after re-running our models) and the sample sizes of the individual birds and observations included.

16. *Lines 186-193 – P values should be accompanied by test scores and df, unless you can find that information elsewhere. Do not see that in Fig. 6 that is referenced.*

We thank the reviewer for this hint. We now report Wald z-statistics for the fixed effects ('same vs different' and 'spring vs autumn') as requested. In negative-binomial GAMs (mgcv), these parametric terms are evaluated using Wald z-tests, which do not have finite degrees of freedom ($df \rightarrow \infty$). This is now also stated in the manuscript text (ll. 211).

17. *L. 209-211 – This is a large study with multiple years of data on the same individuals. However, I would soften this statement some. The terms “large” and “high” are subjective, and there are some good studies out there with solid sample sizes and also have multiple years of data (e.g. Conklin et al. 2013, Léandri-Breton et al. 2021, Kürten et al. 2022). Maybe say that studies like yours are uncommon. If you want to say this study is the first for large numbers and years, I suggest including a table with some of the major studies (or at least reference the currently most significant ones).*

We have followed your suggestion and softened the statement by making use of the suggested studies showing similarly large datasets. Similar to the studies mentioned above, we could not find further studies that assessed repeatability using a large international GPS-tagging dataset (which offers according exact spatial and temporal resolution). Therefore, we have added a sentence to highlight this. As we did not aim to review previous studies, we did not include a table with different studies but rather cited the most relevant ones in the context of our analysis in the text.

18. *L. 220-222 – but this does not necessarily mean that they are not flexible in response to threats. Do you have any specific examples?*

We have now added the examples of mortality during cold spells, constant site use despite ongoing coastal defense construction works in a wintering site of the Wadden Sea as well as the constant site use of breeding curlews in areas of reforestation in western Russia (ll. 249; ll. 367).

19. *I. 235-237 – Seems like you should have the data to test this?*

We agree and have now tested for this: we have created three classes of birds (i.e. short, medium and long-distance migrants) according to the quantiles of the migration distances exhibited by all birds. Subsequently, we have re-scaled the R values of each variable of interest for each season and performed a regression across the three classes. We were not able to find a distinct tendency for short-distance migrants showing different repeatability values than long-distance migrants. This is now stated in the discussion section of the manuscript and discussed in the context with previous studies (II. 264).

20. *I. 238 – what do other studies say about this? This would be a good place to provide some comparison with other studies (examples from birds > Numeniini > curlews). Perhaps put your study in the context of the Franklin et al.'s (2022) review of repeatability and migration phenology?*

Thanks a lot for this valuable suggestion. We agree and have added several sentences to compare other studies, namely the study of Franklin et al. (2022) with our results and in particular with our finding that the R values were always lower for autumn than for spring migration (II. 273).

21. *I. 248 – “...the reason for this behavior is unknown.” – What do other studies suggest?*

We have now cited several studies that point out that a high site-faithfulness of wintering grounds is beneficial in terms of predictive prey resources, lower predation risks and a timely start of postnuptial moult and stated that a repeated arrival at wintering grounds could provide the same benefits (II. 282).

22. *I. 290-295 – Here you say that the curlews show low flexibility to react to environmental change but this seems to be at odds with your discussion above (roughly I. 274-280) where you say that they do not follow the same migration routes and use different stopover sites (and in I. 302 where you say that they are flexible). In the abstract (I. 66) you call curlews a “...stereotypical near threatened species.” Your messaging about whether you think that Eurasian Curlews are flexible or inflexible is inconsistent.*

Please see also our answer to comment no. 8 above. We agree that the discussion on the degree of flexibility in curlews was not consistent enough.

We indeed found that curlews did not follow the exact migration route, however, the routes of the same individuals were significantly more similar to the routes of conspecifics from the same breeding and wintering clusters. This is now clarified in the discussion. As we did not show data / analyze the spatial use of stop-over sites, we have deleted the speculative part on the use of these sites to streamline our discussion.

23. L. 303-305 – Conklin et al. (2021) show that individual Bar-tailed Godwits are able to adjust migration strategies

Thanks a lot for making us aware of this study. We have now added this reference two sentences further above where we are talking also about the potential of black-tailed godwits to adjust migration.

24. L. 308-309 – repeats what you say in l. 214-217.

We agree that this topic was already briefly touched before, i.e. in the context of a comparison between the degree of repeatability of stop-over vs breeding / wintering sites. At the end of the discussion, however, we take up the topic of repeated use of stop-over sites to highlight which follow-up studies are needed. Therefore, we would prefer to keep the discussion of the use of stop-over sites in both places of the discussion.

Comments by reviewer #2: (Ben Lagasse)

1. In general, I liked the manuscript. The length is great, the figures are nice, and the authors have taken care in putting the manuscript together.

Thanks a lot for his positive feedback.

2. There is very little support for the conclusion that the lack of flexibility exhibited by the tracked curlews represents a potentially maladaptive response to environmental change or ecological trap. For example:

#1: For a bird like a Eurasian Curlew, which has high annual survival and a relatively slow life history strategy, high repeatability and prioritizing safety can be a highly adaptive response to environmental change — particularly when facing low predictability and high variability in environmental conditions. For discussions of tradeoffs associated with high repeatability versus flexibility under varying environmental conditions, see:

Lof et al. 2012. *Timing in a fluctuating environment: Environmental variability and asymmetric fitness curves can lead to adaptively mismatched avian reproduction.* <https://doi.org/10.1098/rspb.2012.0431>

Bauer et al. 2020. *Environmental variability, reliability of information and the timing of migration.* <https://doi.org/10.1098/rspb.2020.0622>

We agree that our discussion on a potential maladaptive behavior of curlews needs to be revisited. The main aim of this study was not to relate repeatability to data on environmental change. Therefore, we agree that we are not able to prove if the strong repeatability exhibited by curlews throughout their entire annual cycle is maladaptive per se. In accordance with comment no. 8 made by reviewer #1, we now provide some examples on possible effects of short-term environmental and anthropogenic changes in the landscape of curlews. We agree however, that it is particularly hard to prove effects of long-term changes on curlews. Therefore, we have now extended our critical discussion on the possibility that curlews might adapt to long-term environmental changes also by an altered behavior of future recruits. (II. 377; II. 304) We now present an outlook section on possible next steps that need to be taken in order to solve the question of possible maladaptive behavior of curlews in particular with respect to long-term environmental changes. (II. 385).

We are now also describing the life-history of curlews in more detail and point out tradeoffs associated with high repeatability versus flexibility under varying environmental conditions using the papers suggested. (II. 392).

3. #2: *The manuscript doesn't offer any insight on the extent to which the areas in which Eurasian Curlews bred, migrated, and wintered underwent environmental change. Could the Eurasian Curlews in the study have been capable of surprising flexibility (e.g., lines 261–262) but exhibited high repeatability because of the relative stability of the environments in which they occurred, during the time periods in which they were in them? To conclude that the Eurasian Curlews in the study were potentially maladapted, I think (at a minimum) it would need to be shown that the curlews in the study exhibited*

high repeatability despite significant environmental changes in the locations and time periods in which they were tracked.

We have now included two examples of curlews suffering from short-term environmental changes (i.e. mortality of curlews as a consequence of a cold spell in wintering grounds and effects of afforestation and changes in agricultural practices in northwest Russian breeding sites) and which could have likely been avoided by a higher degree of phenotypic flexibility (II. 367). However, we agree with the reviewer that we are not able to judge if curlews might be maladapted with respect to environmental changes across longer time periods as our tracking study (although comprising seven years of data) is likely too short to draw any conclusions on this. Therefore, we have now clearly pointed out in the discussion that we cannot judge about this point (II. 371) but (in accordance to your comment no. 5) we have now pointed out that conservation management should consider curlews as a species of concern at least for short-term anthropogenic or environmental changes (II. 64; II. 399).

4. #3: *Environmental changes don't necessarily require that individuals alter their migrations. For example, environmental changes during individuals' lifespans may not alter resource distributions to the point that individuals need to flexibly adjust their migrations. Also, individuals with high repeatability in migratory timing may adjust to environmental changes by flexibly adjusting other aspects of their annual routines (Carneiro et al. 2019).*

Carneiro et al. 2019. Why are Whimbrels not advancing their arrival dates into Iceland? Exploring seasonal and sex-specific variation in consistency of individual timing during the annual cycle. <https://doi.org/10.3389/fevo.2019.00248>

We cannot judge if (long-term) environmental changes in our study area were so strong that they altered resource distributions to the point that individuals need to adjust their migrations. However, as you point out and as the study by Carneiro et al. (2019) shows, species encountering changing environments (in this case climate change in high latitudes) may react by keeping several aspects of their annual cycle constant (i.e. timing of spring migration in the case of the cited study), while they adjust other aspects (i.e. timing of breeding and autumn migration). In the case of our study, we found a striking repeatability in all aspects of the annual cycle (departure from breeding grounds which we have explained with different breeding success of the same individuals among years). Therefore, in contrast to the study of Carneiro et al. (2019) our study clearly

highlights the repeatability in every aspect of the annual cycle of curlews. Thus, we have now included Carneiro et al. (2019) (and several other studies) into the discussion section to point out that we do not find indications of varying behavior in any aspects of the annual cycle of curlews (ll. 343). At the same time, we agree that we cannot draw conclusions if environmental changes were so strong that they reached the point to critically alter resource distributions to the degree that an individual needs to react during its life-span. Therefore (also following the conclusion in Carneiro et al. 2019) we have now further emphasized the importance to collect information of variation in the annual cycle or recruits in the course of further generations (i.e. across a longer term; ll. 377; ll. 304).

5. *Some of the above considerations are mentioned in the Discussion but may warrant greater emphasis.*

Done. We have now revisited our discussion section and emphasized the aspects mentioned in your point 2-4 by using the suggested references.

6. *Lines 52–66: Consider (1) reframing the manuscript to explore the extent to which Eurasian Curlews might adapt to environmental changes via processes of phenotypic selection versus phenotypic flexibility, and (2) focusing the take home message of the manuscript on important next steps, conservation implications, or something of the sort.*

Done. We have re-worked the discussion and are now distinguishing between shorter term environmental effects (and provides some examples of missing flexibility) and longer-term environmental effects that our study was not able to capture due to its limited duration of seven years. We now highlight the need for subsequent studies to consider the behavior of different cohorts of curlews to judge the potential of adaptation across generations. As suggested, we have extended our concluding paragraph by an outlook section formulating next steps and implications for conservation of curlews. We have adjusted the last sentences of the abstract accordingly (ll. 64; ll. 399).

7. *I'm thinking Eurasian Curlews molt on their wintering grounds. If so, it could be worth mentioning that here and providing a brief thought on how consistent arrival timing on*

their wintering grounds might (or might not) be tied to individuals' growing high-quality flight feathers in an optimal way. For example, see:

McNamara et al. 1998. The timing of migration within the context of an annual routine. <https://doi.org/10.2307/3677160>.

Barta et al. 2008. Optimal moult strategies in migratory birds. <https://doi.org/10.1098/rstb.2007.2136>

Yes, indeed Eurasian Curlews perform their postnuptial moult within their wintering grounds. So, thanks a lot for this hint. In accordance with comment no. 21 made by reviewer #1, we have now added a small section in the discussion highlighting the potential need of a repeated arrival to wintering grounds to enable onset of the postnuptial feather moult (ll. 282).

8. Lines 308–322: The prior paragraph (lines 290–307) concludes the manuscript. I would remove this paragraph or place it before lines 290–307.

We agree that the second to last paragraph would also be a suitable concluding paragraph, however, given that the original last paragraph was already structured as an outlook section, we have now followed your comment no. 5 and have now used this paragraph to formulate important next steps of research and have drawn conclusions on implications for conservations for curlews.

9. *Comment #4; Lines 427–438: There are multiple aspects of the modeling approach that could be good to address: First, I'm thinking that cluster ID should be fitted as a fixed effect, not a random effect. Unlike modeling exercises in which the fixed effects are the parameters of interest, in repeatability models the random effects are the parameters of interest (Stoffel et al. 2017, page 3). Also, with cluster ID fitted as a fixed effect, mean-centering the data by cluster ID isn't necessary, as there would be a beta estimate per cluster that accounts for differences in means.*

We agree and have revised our analyses accordingly. In all repeatability models, we now treat cluster as a fixed effect and retain individual ID as the random intercept, thereby reporting adjusted repeatabilities (i.e., variability among individuals after accounting for systematic cluster differences; cf. Stoffel et al. 2017). As a consequence, we removed the previous cluster mean-centering step and did not exclude two-individual clusters; cluster-level mean differences are captured by the fixed-effect coefficients.

Re-running all traits with this specification confirmed, and in several cases strengthened, our conclusions. Across parameters and seasons, adjusted

repeatabilities from the fixed-effect cluster models were moderately higher than under the previous centering approach (median $\Delta R \approx 0.12$, maximum ≈ 0.18), without changing the qualitative interpretation (high and significant repeatability throughout). For illustration, spring departure date increased from $R = 0.36$ (previous approach) to $R = 0.54$ (cluster as fixed effect); spring arrival date from $R = 0.40$ to $R = 0.58$. Including sex and capture country as additional fixed effects (see comment no. 3 made by reviewer #1) had negligible influence (median $\Delta R \approx 0.00$). Correlation between repeatabilities from the previous vs. revised models was extremely high (Pearson $r = 0.996$, 95% CI: 0.990–0.998, $p < 0.001$), confirming that the models are consistent and lead to the same qualitative conclusions. We now present the fixed-effect cluster models as our primary analyses and provide the alternative specification (centering) as a sensitivity check (II. 170).

10. *Correctly specifying the model is probably preferable to centering the data by cluster ID (to deal with the non-convergence issue) because it incorporates the hierarchical structure of the data (individuals within clusters) in the model.*

We agree. In the revised analyses we explicitly model cluster as a fixed effect, which directly accounts for hierarchical structure and removes the need for mean-centering (cf., our comment no. 9 above). This specification yielded stable model convergence and confirmed our conclusions.

11. *Second, it is somewhat unclear if the manuscript reports “agreement repeatabilities” or “adjusted repeatabilities” or “enhanced agreement repeatabilities” (Stoffel et al. 2017). As stated, the modeling approach is designed to account for the effect of different GPS fix intervals and cluster IDs, and thus I’m thinking the intention is to report adjusted repeatabilities. Consider clarifying that the manuscript reports adjusted repeatabilities.*

Thank you for this clarification. Our models included log-transformed track duration as a covariate, and in the revised analyses also cluster ID (and in sensitivity analyses sex and capture country) as fixed effects. Following Stoffel et al. (2017), this means that we are reporting adjusted repeatabilities, because the variance explained by fixed effects is excluded from the denominator of repeatability estimates. We have clarified this terminology in the Methods (II. 515) and Results (II. 148).

12. *Third, alongside fitting cluster ID as a fixed effect, it could be worth revisiting the use of Gaussian versus Poisson models. For example, consider using the distribution that describes each migration parameter rather than a one-size-fits-all approach. Poisson distributions are intended to describe counts over time and thus should probably be used to estimate repeatability in total stopover number.*

We agree and have revised the analyses accordingly. We used Poisson GLMMs only for the genuine count variable total rest number, with a pre-specified overdispersion check based on Pearson residuals; where overdispersion was substantial, we applied a squareroot-transformed Gaussian fallback. All other traits are continuous by construction (timing variables such as departure day and arrival day, trip duration, and distances/coordinates) and were analysed with Gaussian models. This trait-specific choice avoids mis-specification and follows standard practice for repeatability estimation. Results from Poisson models for total rest number were consistent with the Gaussian fallback, and the overall conclusions remained unchanged (II. 528).

Thanks a lot for giving us the opportunity to revise our manuscript entitled “High annual-cycle repeatability suggests low flexibility to environmental changes in a near-threatened migratory shorebird” for the second time along the very valuable and constructive suggestions made by the two reviewers.

Please find on the following pages our response (**in bold**) to the questions raised by the two reviewers (*in italics*). Line numbers mentioned by the reviewers refer to the original manuscript version, line numbers stated in our answers refer to the revised version.

Comments by reviewer #1 :

We have followed all the linguistic suggestions made in your word document.

1. *LI. 104: High quality? What do you mean by good?*

We mean that sufficient knowledge is existing. We have rephrased the sentence accordingly.

2. *LI. 131: Still rather vague; > 50%, >75%?*

The Wadden Sea was used by more than half of the tagged curlews. This is now stated.

3. *Fig.1 caption: Include years of study.*

Done.

4. *LI. 165: Should this be “stopovers”?*

We agree and have changed “rests” to “stopover sites” throughout the manuscript (see also our comment no. 7 below).

5. *Fig. 4 caption: All your tables and figures should identify the species you are studying and the years of study.*

Done.

6. *Table 1: Most literature refers to places where the birds stop as “stopover sites”. As Linscott and Senner show, these sites can function more than just “rest” sites. Linscott, J.A. and Senner, N.R., 2021. Beyond refueling: Investigating the diversity of functions of migratory stopover events. *The Condor*, 123(1), p.duaa074.*

We agree and have changed “rests” to “stopover sites” throughout the manuscript. As suggested by reviewer #2, Table 1 was shifted to the Supplement section.

7. *LI. 236: Awkward - What is of interest is that you are analyzing the repeatability of the migration behavior of an animal (Eurasian curlew) using “...an international GPS-tagging dataset...”.*

We agree and have deleted the respective sentence.

8. *LI. 250: Given there are multiple species of curlews, you should be specific in this paper when you are talking about Eurasian curlews vs. other species of curlew*

We have pointed out at the first mention in the text that we refer to Eurasian curlews when we talk about “curlews”.

9. *LI. 305: A previous study of which species? Eurasian curlew?*

We have now stated that this particular study was dealing with black-tailed godwits.

10. *LI. 320: This seems to contradict your statement above l. 289-292*

We agree and have deleted this half sentence.

11. *LI. 333: Now split into two different species - Eurasian and Hudsonian whimbrel (Numenius phaeopus , Numenius hudsonicus)*

Thanks a lot for the hint. We have now provided the scientific names of the new nomenclature.

12. Awkward - suggest - "This would inform conservation..."

We agree and rephrased accordingly.

Comments by reviewer #2:

1. *Line 57: Consider replacing "entire annual cycle" with "entire migratory cycle." The term, entire annual cycle, could include examining nest initiation dates, initiation of molt, completion of molt, etcetera, which were not addressed. Entire migratory cycle seems more precise.*

This is a good point. We have replaced "annual cycle" by "migratory cycle" throughout the entire manuscript when describing or discussing our own data.

2. *Lines 167–175: It's not clear to me why this comparison was needed. It seems that treating wintering and breeding cluster as a fixed effect is the correct specification of the model and there is no need to compare the correctly specified models against mean-centered (i.e., incorrectly specified) models. Consider removing the comparison from the manuscript or elaborating in the Methods (i.e., Lines 534–536) why this comparison was needed.*

We thank the reviewer for this helpful comment. We agree that treating breeding and wintering cluster as a fixed effect represents the appropriate model specification for estimating adjusted repeatabilities. The comparison with the alternative mean-centering approach was originally included to demonstrate the robustness of our results across model specifications. However, we acknowledge that this comparison is not essential for the interpretation of the results and may distract from the primary analysis. We have therefore removed the detailed comparison from the Results section and streamlined the Methods accordingly (ll. 527). The manuscript now focuses exclusively on the fixed-effect specification as the main and conceptually correct approach for estimating adjusted repeatabilities.

3. *Figure 3–5, and Table 1: Consider moving Figure 3–5 and Table 1 to the appendix and replacing them with a single figure that plots the corresponding R values and their 95%*

intervals. Such a figure would need to include corresponding samples sizes, and could distinguish between “significant” and “nonsignificant” results. Doing so would streamline the manuscript without a significant loss of information and would allow rapid and intuitive comparisons of repeatability values between migration parameters. For example, a single figure with R values and their intervals could plainly illustrate 1) the high repeatability of breeding and wintering sites, 2) the higher repeatability of spring migration dates compared to autumn migration dates, and 3) higher stopover repeatability in spring compared to autumn.

We thank the reviewer for this good suggestion. This is indeed a very suitable option to illustrate the main results in a condensed way. Therefore, we have included a new figure with R values and their 95% confidence intervals for both seasons. As suggested, we have now moved Table 1 to the Supplement (Supplement 1, Table S1) to enable the reader to see the concrete R values and their confidence intervals for each parameter which makes a good addition to the newly created figure. However, we feel that the Figures 3-5 where we are plotting the parameters of interest in the first observation year against the consecutive year should stay in the main manuscript text. These figures illustrate the variation of individual curlews for each parameter of interest and thus add an essential part of information to the manuscript. More importantly, they also inform about the absolute values of each parameter of interest. However, should the reviewer or editor feel that the figures 3-5 should be moved into the Supplement section, we would agree to do so.

4. *Lines 266–269: I’m not seeing any mention of this analysis in the Methods or Results. It would be good to clarify that this was a post hoc analysis and present the results in a supplemental figure. Alternatively, the analysis could be included in the Methods and Results.*

This short analysis was included because of a suggestion made by reviewer #1. However, we agree that it should be introduced already in the methods section and the outcome (although non-significant) should be described in the results section. Therefore, we have added respective sentences in the methods (ll. 552) as well as in the results (ll. 182). Furthermore, we are now presenting the results in a supplemental figure (Supplement 3) as suggested.

5. *Lines 299–301: Consider rephrasing this sentence along the lines of, “Although short-term cold spells during the years of our study may have decreased in frequency or severity compared to decades ago (i.e., reference #37), curlews have been described to....”*

Done. Sentence has been rephrased as suggested.

6. *Only observing such behavior in 2 individuals may reflect a lack of flexibility as much as it might reflect a decrease in the frequency and severity of cold spells.*

We have added a respective sentence pointing out that we cannot be sure if the reaction of two individuals might rather reflect missing flexibility or a decrease in the frequency / severity of cold weather events compared to decades ago (ll. 300).

7. *Lines 302–303: I don’t think the results support this statement. At a minimum, it would be good to revise the phrase, “...are less able to...” This manuscript doesn’t really examine what curlews are or aren’t able to do. It describes what curlews happened to do during the years of the study. For example, I think Lines 355–357 do a nice job of describing the results — i.e., “suggests a low flexibility” and “despite this apparent lack of flexibility”.*

We agree that we should not make any comparisons if curlews are adapted to a different degree than gulls (as we are also showing different data than the studies on gull behavior we have cited). Therefore, we have deleted the respective sentence from the discussion.

8. *Lines 304: Again, I think it would be good to revise the phrase, “...be able to...”*

Done. We have rephrased the sentence and deleted the term “be able to” so that the sentence now just describes the fact that some individuals do not react to changes.

9. *Lines 309: “Environmental plasticity” doesn’t seem quite right here. Consider clarifying that the population as a whole might adapt.*

Done. We have rephrased the sentence avoiding the term “environmental plasticity”. We now point out that the population as a whole might adapt.

10. Line 352–355: Consider replacing “entire annual cycle” with “entire migratory cycle.”

The term, entire annual cycle, could include examining nest initiation dates, initiation of molt, completion of molt, etcetera, which were not addressed. Entire migratory cycle seems more precise.

Done. See our comment no. 1 above.

11. Line 399: “phenological plasticity” doesn’t seem quite right here. Consider clarifying

that, “...if curlews lack the flexibility to adapt...”

Done. Sentence was rephrased as suggested.